# Topological Parallax: A Geometric Specification for Deep Perception Models

**Abraham D. Smith**
Geometric Data Analytics, Inc.
343 W. Main Street
Durham, NC 27701 USA
`abraham.smith@geomdata.com`
University of Wisconsin-Stout
Math, Stats, and CS Dept
Menomonie, WI 54751 USA
`smithabr@uwstout.edu`

**Michael J. Catanzaro**
Geometric Data Analytics, Inc.
343 W. Main Street
Durham, NC 27701 USA
`michael.catanzaro@geomdata.com`

**Gabrielle Angeloro**
Geometric Data Analytics, Inc.
343 W. Main Street
Durham, NC 27701 USA
`gabrielle.angeloro@geomdata.com`

**Nirav Patel**
Geometric Data Analytics, Inc.
343 W. Main Street
Durham, NC 27701 USA
`nirav.patel@geomdata.com`

**Paul Bendich**
Geometric Data Analytics, Inc.
343 W. Main Street
Durham, NC 27701 USA
`paul.bendich@geomdata.com`
Duke University
Mathematics Dept
Durham, NC 27708 USA
`bendich@math.duke.edu`

## Abstract

For safety and robustness of AI systems, we introduce *topological parallax* as a theoretical and computational tool that compares a trained model to a reference dataset to determine whether they have similar multiscale geometric structure.

Our proofs and examples show that this geometric similarity between dataset and model is essential to trustworthy interpolation and perturbation, and we conjecture that this new concept will add value to the current debate regarding the unclear relationship between "overfitting" and "generalization" in applications of deep-learning.

In typical DNN applications, an explicit geometric description of the model is impossible, but parallax can estimate topological features (components, cycles, voids, etc.) in the model by examining the effect on the Rips complex of geodesic distortions using the reference dataset. Thus, parallax indicates whether the model shares similar multiscale geometric features with the dataset.

Parallax presents theoretically via topological data analysis [TDA] as a bi-filtered persistence module, and the key properties of this module are stable under perturbation of the reference dataset.

# 1    Introduction

Suppose $X$ is a finite subset of $V = \mathbb{R}^n$ with the Euclidean metric.[1] In data science—particularly in applications of DNNs—we often encounter the situation where $X$ is a dataset, and some opaque algorithm has produced a trained model $k : V \to \{0, 1\}$, where $k(x) = 1$ for all $x \in X$. This defines the model as a set of accepted inputs $K = \{x : k(x) = (1)\} \subset V$, which has no available description beyond evaluation of the perception function $k$ on samples.

Our main contribution in this paper is a method we call *topological parallax* to estimate the multiscale geometry of $K$ from the persistent homology ([14, 37]) of $X$, in a situation where $K$ does not have an explicit description.[2] This method provides meaningful geometric information about $K$ through a simple computational approach that can be applied to any perception model $k$. We prove that the resulting criterion of *homological matching* satisfies a stability property. We propose homological matching via parallax as a geometric specification that could be applied to many machine-learning systems. The measurement of homological matching also admits a back-propagation scheme, which could be used to improve the geometric similarity between the model $K$ and the dataset $X$.

Because of the generality of Definition 1.1, it may be that $V$ represents any layer of a neural network, $X$ represents any dataset mapped into that layer, and $K$ represents an activated region in that layer. For example, the "neural collapse" concept from Papyan et al. [29] can be seen as a special case of this specification, because the conception from [29] is that the dataset $X$ becomes a tight blob in the penultimate layer, and the penultimate model is simply a Voronoi region $K$ surrounding that blob.

## 1.1    Assumptions and Motivation

As discussed by Belkin [5], DNNs usually achieve high statistical accuracy, but some resulting models are better than others at capturing patterns in the dataset. Despite having perfect statistical accuracy on the dataset $X$, fundamental questions arise about the model $K$: "Is it safe to deploy? Is it trustworthy? Is it a good model?" To broadly paraphrase [5], it used to be good practice to tell data analysts not to overfit their data, because overfit models were "bad" due to poor generalization; however, essentially every DNN fits the training data perfectly, so it is not clear what distinguishes "good" models from "bad." *We suggest that a model $K$ is "good" if the geometry of $K$ matches the geometry of $X$.* Consider the two example models in Figure 1. Although both models achieve perfect statistical accuracy, only one of them appears to have learned the geometric structure of the dataset. This suggestion is likely intuitive to many ML engineers, but the subtlety lies in the implicit assumptions often made about the nature of the dataset and the available models. We assess this matching in a way that is independent of the architecture that produced $k$, and that makes very few assumptions about $X$ and $K$, as encapsulated by Definition 1.1, which are assumed henceforth.

**Definition 1.1** (Datasets and Models). *Suppose that $V$ is a geodesic space. We define a model $K \subset V$ to be the closure of an open set, $K = \overline{K^\circ} \subset V$, colloquially known as a "solid." For any finite dataset $X \subset V$, we consider the collection of all models for which $X$ is contained in the interior of the model, $X \subset K^\circ$; let $\mathcal{M}(X) \coloneqq \{K \subset V : X \subset K^\circ, \overline{K^\circ} = K\}$. With subset inclusions as morphisms, $\mathcal{M}(X)$ is a small category. For any $K \subset V$ with $K = \overline{K^\circ}$, let $\mathcal{M}^*(K) \coloneqq \{X \subset V : X \subset K^\circ, \#X < \infty\}$. With functions of finite sets as morphisms, $\mathcal{M}^*(K)$ is a small category. Of course, $X \in \mathcal{M}^*(K) \Leftrightarrow K \in \mathcal{M}(X)$. Note that $V \in \mathcal{M}(X)$.*

Note that we make no assumptions whatsoever about the architecture or training method that yielded $K$. With such a broad definition of $X$ and $K$, we need a notion of "geometry" with very few assumptions. We use standard topological notation [23, 17, 14].

**Definition 1.2** (Void). *Given a set $K \subset V$, an* interpolative void *in $K$ is a bounded open set in the complement of $K$, $\Omega \subset K^c$, such that there exists a pair of points $x, y \in K$ for which all $V$-geodesics pass through $\Omega$. If $V = \mathbb{R}^n$, this means $\Omega$ intersects the convex hull of $K$.*

---

[1]The theoretical results apply if $V$ is any geodesic space—a metric space in which each distance is realized by a path—but our motivation and examples use $V = \mathbb{R}^n$ to avoid distraction from our central theme of model assessment.

[2]Topological parallax was named by analogy to the method in astronomy, where an inaccessible object cannot be measured directly (in this case, the geometry of $K$), so we must infer its location by comparing multiple observations from available vantage points (in this case, the dataset $X$).

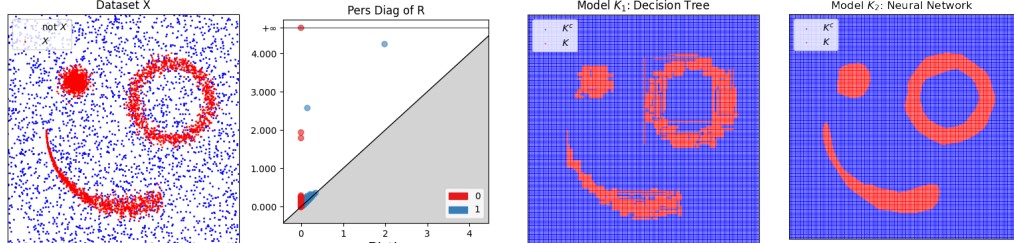

Figure 1: 1: A dataset $X \subset \mathbb{R}^2$. 2: The persistence diagram of the Rips complex of $X$, showing three components (two of which die at the dim-0 dots near 2.0) and two cycles (the dim-1 dots, representing the annulus and the overall arrangement) among the persistent features. 3: A tree model $K_1$. 4: A neural network model $K_2$. The tree model has many small voids that would forbid interpolation or perturbation in many locations. Without the luxury to visualize models $K \subset V$ for $\dim V > 3$, Parallax can measure geometric similarity between $X$ and $K$ via the Rips complex $R$ of $X$.



Figure 2: Five models for the same dataset, and the scales detected by parallax. The circle have radii 20 and 30, respectively. See Section 6 for general interpretation. The missing edge causing $\lambda_{\mathrm{sup}}$ is highlighted in red at filtration radius 7.1, and the missing edges that cause $\lambda_{\mathrm{hi}}$ are highlighted in blue. The dashed blue edge does not affect $\lambda_{\mathrm{hi}}$ because the larger cycle determines that value.

Why focus on voids? As observed by Balestriero et al. [2], when $V$ is high-dimensional, it is unlikely that any points in $X$ lie in the convex hull of any others. However, it also seems unlikely that a "good" model $K$ of $X$ will be merely a convex solid. If convex models were sufficient for real-world problems, DNNs would be unnecessary, and the field would have concluded with PCA, Gaussian kernels, and convex polytopes. For example, many models implicitly or explicitly rely on the so-called manifold hypothesis, which is the hope that realistic datasets $X$ will tend to be distributed near a union of lower-dimensional manifolds immersed in $V$, in which case a "good" model $K$ would be a slight thickening of those manifolds to allow for measurement error. If there are multiple manifolds or there is nonzero curvature, such a model will have voids.

Any void in $K$ indicates a region where $K$ does not allow interpolation, which is where interesting geometry occurs. Voids occur if and only if some geodesic in $K$ is strictly longer than the corresponding geodesic in $V$. Hence, for this article, we interpret "geometry of $K$" as the presence or absence of voids. We do not make assumptions about the geometric features of $X$ or about the family from which $K$ is chosen; rather, we ask that $K$ respects the features of $X$, whatever they are. We propose that a "good" model $K$ is one whose voids represent the highly persistent features of $X$, in the sense of Topological Data Analysis [TDA, [14, 27]]

A model $K$ can be "bad" because it has too many or too few voids at various scales. For example, mismatch of voids of $K$ and features of $X$ would indicate that $K$ is over-sensitive to small error or under-sensitive to large error, either of which could lead to adversarial attack. Also, numerous small-scale voids could make $K$ incompatible with some forms of the manifold hypotheses, by obstructing the coverage of $K$ by an atlas of local convex charts of moderate size.

## 1.2 Outline

Section 2 introduces our key object, the bi-graded [7] parallax complex, in Definition 2.2, which measures geodesic distortion via the Rips complex. Section 3 provides a notion of dataset perturbation

and shows that the parallax complex and its homology remain stable under those perturbations. Section 4 defines *local simplicial matching* and shows how parallax detects small-scale changes in the Rips complex of $X$ in $K$ versus $V$, giving a clearly interpretable scale of locality, $\lambda_{\text{lo}}$. Section 5 defines *homological matching* and shows how parallax detects large-scale voids in $K$, giving a scale $\lambda_{\text{hi}}$ above which homological features in $X$ are respected by voids in $K$. Together, these results provide an overall interpretation as a specification in Section 6, which largely achieves the goals laid out in Section 1. Section 7 provides computational approaches to computing parallax, and links to our open-source software that has many practical improvements not detailed in this paper. Section 8 illustrates the effectiveness of parallax as a specification, as demonstrated on two models using the cyclo-octane dataset [22]. Additional proofs, details, and examples are provided in the Supplementary Material appendices.

## 1.3 Related Work

To the best of our knowledge this is the first work to use TDA to express a desired geometric relationship that holds directly between datasets and models trained on them. There has been some work, for example the Manifold Topology Divergence of Barannikov et al. [3] or the Geometric Score of Khrulkov and Oseledets [20], which uses various TDA-based measures to quantify the difference between training data and new data generated by generative models. Quite a few other papers (see Fernández et al. [15] for a very recent example) use TDA-based constructions to infer properties of underlying data manifolds, usually under very strict sampling assumptions.

More broadly, there has been a recent explosion of work (e.g. Hensel et al. [19]) connecting TDA to ML/DL. Several works (e.g. Adams et al. [1], Bendich et al. [6]) use TDA as a *feature extraction* method, pre-processing more complicated data objects before running standard ML pipelines. Later works (e.g. Chen et al. [10], Demir et al. [12], Solomon et al. [32], Nigmetov and Morozov [25]) use TDA to define novel losses within ML algorithms. Note that we comment below on ways in which our notion of homological matching can be used to define a TDA-based loss. TDA has also been used (e.g. Naitzat et al. [24], Wheeler et al. [36]) to analyze the behavior of data as it passes through the layers of a DNN. Some works (e.g. Guss and Salakhutdinov [16]) assess the capacity of a specific DNN to classify datasets with specific shapes, but do not provide tools to quantify shape mismatch between model and dataset. Finally, several works (e.g. Carriere et al. [8], Papillon et al. [28]) use TDA to define novel DNN architectures, including GNNs and other higher-order combinatorial structures.

There is also a recent stream (e.g. Liu et al. [21], Wang et al. [35]) of work that builds validation and verification/falsification techniques for desired properties of DNN-trained models; these mostly focus on the mechanics of how to verify/falsify such properties, rather than attempting to define them as we do. Perhaps the closest work in this stream to ours is Dola et al. [13], which uses a prior assumption on the underlying data distribution to verify/falsify DNN properties.

## 2 The Parallax Bi-Complex

Let $B_\alpha(x)$ denotes the closed geodesic ball of radius $\alpha$ around $x \in V$. For a formal edge $e = (x_0, x_1)$ between points in $X$, $\rho_V(e)$ is the minimum radius for which $B_{\rho_V(e)}(x_0)$ intersects $B_{\rho_V(e)}(x_1)$. Thus, $2\rho_V(e)$ is the geodesic distance between $x_0$ and $x_1$. The Rips complex $R(X, V)$ is the simplicial complex generated by these edges, as filtered by $\rho_V(e)$. A chain is a formal sum of simplices in a complex [14]. More generally, for any $K \in \mathcal{M}(X)$, the Rips complex $R(X, K)$ and its filtration by $0 \le \alpha < \infty$ is defined by

$$(x_0, \ldots, x_d) \in R_d(X, K)_\alpha \text{ if and only if } B_\alpha(x_i) \cap B_\alpha(x_j) \cap K \ne \emptyset, \ \forall \ 0 \le i < j \le d. \quad (2.1)$$

For any chain $Y \in R(X, K)$, let $\rho_K(Y) = \min\{\alpha : Y \in R(X, K)_\alpha\}$. When $X$ and $V$ are understood in context, we abbreviate $R = R(X, V)$ for the ambient case $K = V$.

**Lemma 2.1.** *If $K_1, K_2 \in \mathcal{M}(X)$ with $K_1 \subset K_2 \subset V$, then there is a natural inclusion of filtered modules, $R(X, K_1)_\alpha \subset R(X, K_2)_\alpha \subset R_\alpha$. That is, $\rho_V(Y) \le \rho_{K_2}(Y) \le \rho_{K_1}(Y)$, with the convention $\min \emptyset = \infty$.*

The previous lemma is simply because geodesic lengths in $K_1$ are never shorter than geodesic lengths in $K_2$, and neither is shorter than geodesic lengths in $V$. Our approach to the question "does the geometry of K match the geometry of X?" relies on detecting the inequality $\rho_V(Y) < \rho_K(Y)$.

**Definition 2.2** (Parallax Complex). *For $K \in \mathcal{M}(X)$, let $P(X, K, V)$ denote the subcomplex of $R$ defined for each real pair $(\alpha, \varepsilon)$ by*

$$P(X, K, V)_{\alpha, \varepsilon} = \{Y \in R \; : \; \rho_K(Y) \leq \alpha, \; \rho_V(Y) \leq \rho_K(Y) \leq \rho_V(Y) + \varepsilon\}. \qquad (2.2)$$

*When $X, K, V$ are understood in context, we abbreviate $P = P(X, K, V)$.*

The parameter $\varepsilon$ measures the distortion of geodesic length in $K$ versus $V$. The next few lemmas are immediate consequences of the definition.

**Lemma 2.3** (Parallax is Bi-Filtered). *If $\alpha \leq \alpha'$, then $P_{\alpha, \varepsilon} \subset P_{\alpha', \varepsilon}$. If $\varepsilon < \varepsilon'$, then $P_{\alpha, \varepsilon} \subset P_{\alpha, \varepsilon'}$.*

**Lemma 2.4.** *For all $\alpha, \varepsilon$, we have $P_{\alpha, \varepsilon} \subset R(X, K)_\alpha \subset R_\alpha$.*

Let $\iota : P_{\alpha, \varepsilon} \to R_\alpha$ denote the inclusion of complexes, and let $\iota_* : HP_{\alpha, \varepsilon} \to HR_\alpha$ denote the induced homomorphism on homology [17].

**Corollary 2.5** (Homology Deaths are later in Parallax). *Suppose that $[Y]$ is a class in $HP_{\alpha_1, \varepsilon_1}$ such that the bi-transition map $HP_{\alpha_1, \varepsilon_1} \to HP_{\alpha_2, \varepsilon_2}$ takes $[Y] \mapsto [0]$. Then there exists $t \leq \alpha_2$ such that the transition map $HR_{\alpha_1} \to HR_t$ satisfies $\iota_*([Y]) \mapsto [0]$.*

**Lemma 2.6.** *For all $\varepsilon \geq \alpha$, we have $P_{\alpha, \varepsilon} = P_{\alpha, \infty} = R(X, K)_\alpha \subset R_\alpha$.*

The proofs of 2.5 and 2.6 are given in the Supplemental Material.

It is sometimes useful to create single-parameter filtrations through $P$, parameterized by Rips radius, for the purpose of computing barcodes and persistence diagrams.

**Definition 2.7** (Rips-like Path). *A Rips-like path is a filtered module $L_\alpha = P_{\alpha, \varepsilon(\alpha)}$ for $0 \leq \alpha < \infty$ such that $\varepsilon(\alpha)$ is a non-decreasing function satisfying $\varepsilon(0) = 0$.*

A Rips-like path has homology $HL$ and a barcode or persistence diagram. By Lemma 2.6, one Rips-like path is $R(X, K)_\alpha = P_{\alpha, \alpha}$. Another is the "inflexible" path $L_\alpha = P_{\alpha, 0}$.

## 3 Perturbation

This section establishes lemmas that ensure Parallax and its consequences (notably Theorem 5.4) are stable under certain types of perturbations, which means that the parallax is reasonable in the presence of noise.

**Definition 3.1** (Pointwise Perturbation). *Given $X \in \mathcal{M}^*(V)$, a pointwise $\kappa$-perturbation is $X' \in \mathcal{M}^*(V)$ such that the sets $X$ and $X'$ admit a one-to-one correspondence $f : X \to X'$ satisfying $\|f(x) - x\| \leq \kappa$. We write $f : X \overset{\kappa}{\approx} X'$.*

**Definition 3.2** (Pointwise $K$-perturbation). *Suppose $f : X \overset{\kappa}{\approx} X'$ such that each $(x, f(x))$ pair is connected by a $K$-geodesic of length $\leq \kappa$. We write $f : X \overset{\kappa}{\approx}_K X'$.*

Note that any $f : X \overset{\kappa}{\approx} X'$ is an isomorphism in the category $\mathcal{M}^*(V)$, and any $f : X \overset{\kappa}{\approx}_K X'$ is an isomorphism in the category $\mathcal{M}^*(K)$. The relation $\overset{\kappa}{\approx}$ is reflexive and symmetric, but not transitive; hence, it is a way of describing proximity but does not provide an equivalence relation. Of course, $X' \overset{\kappa}{\approx} X$ implies the Hausdorff distance satisfies $d_H(X, X') \leq \kappa$. A pointwise $\kappa$-perturbation can cause length distortions by $2\kappa$, as said formally in the following lemma.

**Lemma 3.3** (Data Perturbation Lemma). *If $f : X \overset{\kappa}{\approx} X'$, then the Rips complexes identified via $f$ admit a $\kappa$-interleaving*

$$\cdots \longrightarrow R(X', V)_{\alpha - \kappa} \overset{f_\sharp^{-1}}{\longrightarrow} R(X, V)_\alpha \overset{f_\sharp}{\longrightarrow} R(X', V)_{\alpha + \kappa} \overset{f_\sharp^{-1}}{\longrightarrow} R(X, V)_{\alpha + 2\kappa} \longrightarrow \cdots$$

*Specifically, for any edge $e = (x_i, x_j) \in R(X, V)_\alpha$ defined by the existence of a $V$-geodesic of length $2\alpha$, the edge $f_\sharp(e) = (f(x_i), f(x_j))$ in $R(X', V)$ has length $2\alpha'$ satisfying $2\alpha - 2\kappa \leq 2\alpha' \leq 2\alpha + 2\kappa$.*

*Moreover, the same holds when replacing $V$ with $K$, under the assumption $X \overset{\kappa}{\approx}_K X'$.*

*Proof.* Note that a geodesic of length $2\alpha$ corresponds with the intersection of two balls of radius $\alpha$; hence, the factor of 2. The worst-case perturbation is to move each of $x_i$ and $x_j$ by $\kappa$ in opposite directions, away from each-other, along their geodesic. $\qquad\square$

**Lemma 3.4** (Parallax Interleaving Lemma). *Suppose $f : X \overset{\kappa}{\approx}_K X'$. Let $P = P(X, K, V)$ and $P' = P(X', K, V)$. For any $\alpha, \varepsilon$, these parallax complexes admit a $(\kappa, 2\kappa)$-interleaving*

$$\cdots \longrightarrow P_{\alpha,\varepsilon} \xrightarrow{\;f_\sharp\;} P'_{\alpha+\kappa,\varepsilon+2\kappa} \xrightarrow{\;f_\sharp^{-1}\;} P_{\alpha+2\kappa,\varepsilon+4\kappa} \longrightarrow \cdots$$

**Corollary 3.5** (Parallax Interleaving Lemma, Homology Version). *Suppose $f : X \overset{\kappa}{\approx}_K X'$. Let $HP = HP(X, K, V)$ and $HP' = HP(X', K, V)$. For any $\alpha, \varepsilon$, these homology groups admit a*

$(\kappa, 2\kappa)$*-interleaving* $\cdots \longrightarrow HP_{\alpha,\epsilon} \xrightarrow{\;f_*\;} HP'_{\alpha+\kappa,\epsilon+2\kappa} \xrightarrow{\;(f^{-1})_*\;} HP_{\alpha+2\kappa,\epsilon+4\kappa} \xrightarrow{\;f_*\;} \cdots$

The proof of Lemma 3.4 is a worst-case distance estimate given in the Supplemental Material, and Corollary 3.5 follows functorially.

# 4 Sampling Density and Local Simplicial Matching

The goal "the geometry of $K$ should match the geometry of $X$" requires that $X$ has sufficient sampling density throughout $K$ to express a meaningful comparison. The ideal situation would require there is a (small) scale $\lambda$ for which: (1) $K^\circ$ is homeomorphic to $\bigcup_{x \in X} B_\lambda(x)$, so that these $\lambda$ balls capture the topology of $K$; (2) all "highly persistent" homological features of $X$ are born before $\lambda$; and (3) $\bigcup_{x \in X} B_\lambda(x) \subset K$, so that perturbations of size $\lambda$ in the dataset $X$ are allowed, and so that these balls can be used as local charts in $K$. These sampling properties may or may not be true for any particular pair $(X, K)$, but Definition 4.1 provides scales for comparison.

**Definition 4.1** (Locally Simplicially Matched). *We say that $X$ and $K \in \mathcal{M}(X)$ are $\lambda$-locally simplicially matched [LSM] if the subset $P_{t,0} \subset R_t$ is an equality for all $t \leq \lambda$. For any $K \in \mathcal{M}(X)$, the first non-LSM scale realized by the Rips complex is*

$$\lambda_{\mathrm{sup},X}(K) := \sup\{\lambda \,:\, X, K \text{ are } \lambda\text{-LSM}\} = \min\{\rho_V(Y) \,:\, \rho_K(Y) > \rho_V(Y) \,\exists Y \in R\}.$$

*The last LSM scale realized by the Rips complex is*

$$\lambda_{\mathrm{lo},X}(K) := \max\{\lambda < \lambda_{\mathrm{sup},X}(K) \,:\, \rho_V(Y) = \lambda \,\exists Y \in R\}.$$

*Another LSM scale is $\lambda_{\mathrm{ball},X}(K) := \max\{\lambda \,:\, \bigcup_{x \in X} B_\lambda(x) \subset K^\circ\}$.*

When $X$ and $K$ are $\lambda$-LSM, we will identify $P_{t,0}$ with $R_t$ so that $H_* P_{t,0} = H_* R_t$ whenever $t \leq \lambda$. Furthermore, locally simplicially matched implies $P_{t,\epsilon} = R_t$ for any $\epsilon > 0$ and $t < \lambda_{\mathrm{sup},X}(K)$, following directly from the definition of $P_{a,\epsilon}$.

**Lemma 4.2.** *If $K \in \mathcal{M}(X)$, then $0 < \lambda_{\mathrm{ball},X}(K) < \lambda_{\mathrm{sup},X}(K)$.*

However, it may be that $\lambda_{\mathrm{lo},X}(K) = 0$, if the shortest edge $e \in R$ has $\rho_V(e) < \rho_K(e)$.

**Corollary 4.3.** *For any $K, K' \in \mathcal{M}(X)$, there is some $0 < \lambda$ such that each pair $(X, K)$ and $(X, K')$ is $\lambda$-locally simplicially matched.*

**Corollary 4.4.** *For any $X, X' \in \mathcal{M}^*(K)$, there is some $0 < \lambda$ such that each pair $(X, K)$ and $(X', K)$ is $\lambda$-locally simplicially matched.*

**Lemma 4.5.** *If $K_1, K_2 \in \mathcal{M}(X)$ and $K_1 \subset K_2$, then $\lambda_{\mathrm{sup},X}(K) \leq \lambda_{\mathrm{sup},X}(K')$.*

The next lemma provides a bound on $\lambda_{\mathrm{sup}}$ under small data perturbations, which is a form of stability.

**Lemma 4.6.** *Assume Euclidean $V$. Suppose that $K \in \mathcal{M}(X) \cap \mathcal{M}(X')$. If $f : X \overset{\kappa}{\approx}_K X'$ for $\kappa < \frac{1}{2}\lambda_{\mathrm{ball},X}(K)$ then $\frac{1}{2}\lambda_{\mathrm{ball},X}(K) - \kappa \leq \lambda_{\mathrm{sup},X'}(K)$.*

The proof of Lemma 4.6 is a triangle-inequality argument in the Supplemental Material. The other results are immediate observations from the definitions.

# 5 Homological Matching

From Section 1, our purpose is to determine whether the geometry of $K$ matches the geometry of $X$. In Section 4, we introduced "local simplicial matching" as a way to compare small-scale geometry. In this section, we introduce "homological matching" as way to compare large-scale geometry. Generally this is done by asking whether highly persistent features of $X$ in $V$ (as measured by $HR$) are also highly persistent as features of $X$ in $K$ (as measured by $HP$). This comparison is sensible if $X$ and $K$ are $\lambda$-locally simplicially matched, so that cycles can be identified between $HR_\lambda = HP_{\lambda,0}$. We phrase it algebraically in Definition 5.1, but Lemma 5.3 provides the interpretation that, among cycles born before $\lambda$, those of long persistence $(\delta - \lambda)$ in $HR$ have even longer persistence $(\omega - \lambda)$ in $HL$, meaning that $K$ has large-scale homological features corresponding to those of $X$.

**Definition 5.1** (Homologically Matched). *For $K \in \mathcal{M}(X)$, and $0 \leq \lambda < \lambda_{\sup,X}(K) < \delta \leq \omega$, we say that $X$ and $K$ are $(\lambda,\delta,\omega)$-homologically matched [HM] if the transition maps of $HR$ and $HP$ satisfy* $\ker HR_{\lambda \to \delta} \subset \ker HP_{(\lambda,\varepsilon_1) \to (\omega,\varepsilon_2)}$ *for some $0 \leq \varepsilon_1 \leq \varepsilon_2$. Equivalently, if* $\ker HR_{\lambda \to \delta} \subset \ker HL_{\lambda \to \omega}$ *for some Rips-like path $L$.*

Definition 5.1 is guided by the Void Lemma (5.2) and the Matching Lemma (5.3).

**Lemma 5.2** (Void Lemma). *Suppose $K \in \mathcal{M}(X)$ and $0 < \lambda_{lo,X}(K)$. Let $L$ be any Rips-like path. If $[Y] \in HR(X,V)$ with birth $b < \lambda_{lo,X}(K)$, then the deaths $c$, $d,e$ of $[Y]$ in $HR(X,V)$, $HR(X,K)$, $HL$, respectively, satisfy $c \leq d \leq e$. Moreover, $c < e$ implies that $K$ has a void that disrupts the death of $[Y]$, and that void contains a ball of radius $r$ satisfying $(\pi - 2)r \leq 2(d - c) \leq 2(e - c)$.*

In particular, if the class $[Y]$ has $e = \infty$ for the "inflexible" path $L_t = P_{t,0}$, then $K$ contains a void. The proof of Lemma 5.2 appears in the Supplementary Material and uses simple distance estimates.

**Lemma 5.3** (Matching Lemma). *Suppose that all pairs in $X$ have distinct lengths, and that $X$ and $K \in \mathcal{M}(X)$ are $(\lambda,\delta,\omega)$-HM. Then there is a Rips-like path $L$ for which each dot $(b,d)$ in the persistence diagram of $L$ (or bar in the barcode) with $b \leq \lambda < \omega < d$ corresponds via $\lambda$-LSM to a dot $(b,c)$ in the persistence diagram of $R$ with $b \leq \lambda < \delta < c$.*

Note: to check whether $X$ and $K$ are $(\lambda,\delta,\omega)$-HM, it suffices to check a Rips-like path through $P_{\lambda,0}$ and $P_{\omega,\infty}$. At the other extreme, we get an overly-strict bound by checking the "inflexible" Rips-like path $L_t = P_{t,0}$.

Because of the filtration stability of $HP$ in Corollary 3.5, $(\lambda,\delta,\omega)$-HM is also stable to perturbation in $X$, as seen in Theorem 5.4.

**Theorem 5.4** (Stability of Homological Matching). *Suppose that $X$ and $K$ are $(\lambda,\delta,\omega)$-HM. If $f : X \overset{\kappa}{\approx}_K X'$ such that $X'$ and $K$ are $(\lambda - \kappa)$-LSM, then $(X',K)$ are $(\lambda - \kappa, \delta - \kappa, \omega + \kappa)$-homologically matched.*

The proof of Lemma 5.3 is functorial, and Theorem 5.4 is a diagram chase. Both are given in the Supplemental Material.

**Definition 5.5.** *Given $X$ and $K \in \mathcal{M}(X)$ let $\lambda_{hi,X}(K)$ denote the minimum of those $\delta$ for which $X, K$ are $(\lambda_{lo,X}(K), \delta, \infty)$-HM.*

# 6 Interpretation and Specification

Therefore, to answer our original purpose, we can assess whether a model $K \in \mathcal{M}(X)$ is a "good geometric match" for $X$ using the following procedure: (1) Regardless of $K$, examine the persistence diagram of $R(X,V)$ to identify dots with early birth and long persistence, which TDA theory tells us (e.g. the Homology Inference Theorem [11]) should indicate genuine geometric features of $X$; (2) for a model $K \in \mathcal{M}(X)$, compute $\lambda_{lo,X}(K)$ and $\lambda_{hi,X}(K)$; and (3) check whether those dots are born before $\lambda_{lo,X}(K)$ and die after $\lambda_{hi,X}$.

If we believe that the multiscale geometric patterns among the points in $X$ is meaningful, then this procedure is essentially a specification for how a "good" model ought to behave.

This process can give a "bad geometric match" in various ways, for example: if step (1) does not show a clear collection of well-separated dots, then it is unlikely that $X$ actually has computable geometry that can be captured with the Rips complex; if step (2) yields $\lambda_{lo,X}(K) = 0$, then $K$ has

voids between every pair of points in $X$, possibly due to under-sampling or over-fitting, and should not be trusted for any interpolative purpose; or if step (3) shows that the quadrant to the upper-left of $(\lambda_{\text{lo},X}(K), \lambda_{\text{hi},X}(K))$ in the birth-death plane does not capture the desired dots of $X$, then $K$ is failing to capture specific high-persistence features of $X$.

# 7 Computational Methods

In this section, we provide algorithms to estimate $P_{\alpha,\varepsilon}$ in the practical case $V = \mathbb{R}^n$. These algorithms are implemented in Python (and development continues) in our open-source software at https://gitlab.com/geomdata/topological-parallax

The Rips complex $R$ can be computed efficiently, using [26, 33]. But, the set $K$ is known only through the indicator function $k$, and the Rips complex $R(X, K)$ cannot be computed directly.

Consider $x, y \in X$ joined by a $V$-geodesic (line segment) $\overline{xy}$ of length $2\rho_V(e)$, representing a Rips edge $e \in R(X, V)_{\rho_V(e)}$. We would like to estimate $\rho_K(e)$, thus giving $e \in P_{\alpha,\varepsilon}$ for $\alpha = \rho_K(e)$ and $\varepsilon = \alpha - \rho_V(e)$. Some simple geometric observations allow us to estimate $\alpha$ and $\varepsilon$.

**Lemma 7.1.** *If there exists $p \in \overline{xy}$ such that $k(p) = 0$, then $e \notin P_{\alpha,0}$.*

This is because in $\mathbb{R}^n$, $\overline{xy}$ is the only path of minimal length.

**Algorithm 7.2** (Estimation of $e \in P_{\alpha,0}$)**.** *For each $e \in R_\alpha$, sample points $p \sim \overline{xy}$ along the corresponding line segment $\overline{xy}$. (One method of sampling is simply to check the barycenter.) Return True for $e$ if and only if "$k(p) = 1 \forall p \sim \overline{xy}$"*

**Definition 7.3** (Transverse Disk)**.** *Let $V = \mathbb{R}^n$. Given an edge $e \in R$ and a radius $r$, let $D_r(e)$ denote the codimension-1 disk, oriented perpendicular to $\overline{xy}$, and centered at $e$'s barycenter $\frac{x+y}{2}$.*

Any continuous path from $x$ to $y$ that does *not* intersect $D_r(e)$, must have length exceeding $2\sqrt{(\rho_V(e))^2 + r^2}$. If $k(B_r(e)) = \{0\}$, then all $K$-paths avoid $D_r(e)$, so all $K$-paths representing $e$ must have length exceeding $2\sqrt{(\rho_V(e))^2 + r^2}$, giving the following Lemma.

**Lemma 7.4.** *If $K \cap D_r(e) = \emptyset$, then $\sqrt{(\rho_V(e))^2 + r^2} < \rho_K(e)$ and $\frac{1}{2\rho_v(e)} r^2 < \rho_K(e) - \rho_V(e)$.*

For the second inequality, recall the 2nd order Taylor approximation $\rho_V(e) + \frac{1}{2\rho_V(e)} r^2 \leq \sqrt{(\rho_V(e))^2 + r^2}$.

**Algorithm 7.5** (Bounding $e \in P_{\alpha,\varepsilon}$)**.** *For each $e \in R(X, V)$, From $r = 0$, loop:*

  1. *Evaluate $k(p)$ for samples $p \sim D_r(e)$.*

  2. *If $k(p) = 0 \,\forall p$, increment $r$. Else, break.*

*Return the lower bounds $\sqrt{(\rho_V(e))^2 + r^2} \leq \alpha$ and $\alpha - \rho_V(e) = \varepsilon$.*

These algorithms can be extended easily to sample radii along a sequence of points on the edge $e$, thus providing an estimated $K$-path for $e$.

## 7.1 Back-Propagation

Recent work has shown that various topological properties can be expressed as loss functions that are compatible with back-propagation methods; for example [9, 30, 32]. The method in [32] allows back-propagation for a piecewise-smooth loss function of the form $\Phi(\text{PDiag}(f))$, where $f$ is the filtration function on a simplicial complex, and $\text{PDiag}(f)$ is the persistence diagram for the $f$-persistent homology of that complex. Lemma 5.3 allows us to interpret homological matching via persistence diagrams $\text{PDiag}(f(Y))$, where $f(Y) = \min\{t : Y \in L_t\}$ for some Rips-like path $L$.

The following function $\Phi$ could be used to improve homological matching in this framework. Suppose that $X$ and $K$ are $\lambda_{\text{lo}}$-LSM. Consider the persistence diagrams $\text{PDiag}(R)$ and $\text{PDiag}(L)$, where $L$ is some Rips-like path through $P(X, K)$. By $\lambda_{\text{lo}}$-LSM, we know that $\text{PDiag}(R)$ and $\text{PDiag}(L)$ are identical to the lower-left of $(\lambda_{\text{lo}}, \lambda_{\text{lo}})$. Choose a desired target value of $\lambda_{\text{hi}}$. Now, we alter the filtration on $L$ by the following $\Phi$: for dots $(x, y)$ to the lower-left of $\text{PDiag}(L)$,

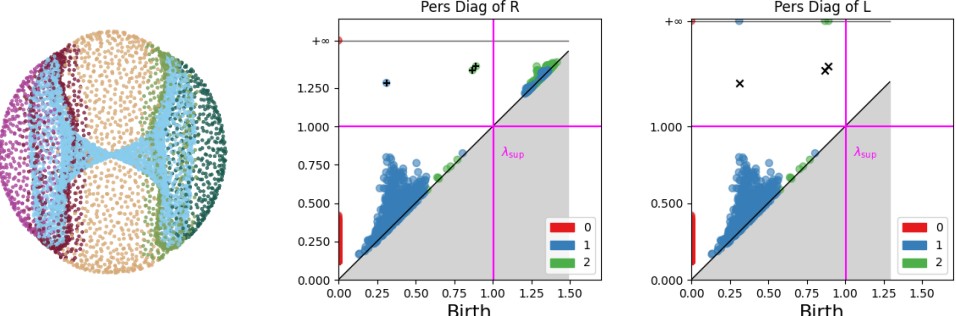

Figure 3: (1) Valid cyclo-octane data under 3D Isomap, colored for viewability. (2) Persistence Diagram of valid data, showing two 2-cycles and one 1-cycle. (3) Persistence Diagram from Parallax of model $K_2$, showing $\lambda_{\text{sup}}$ in magenta and three homologically matched cycles moved to infinity. These diagrams use diameter, not radius, via gudhi.

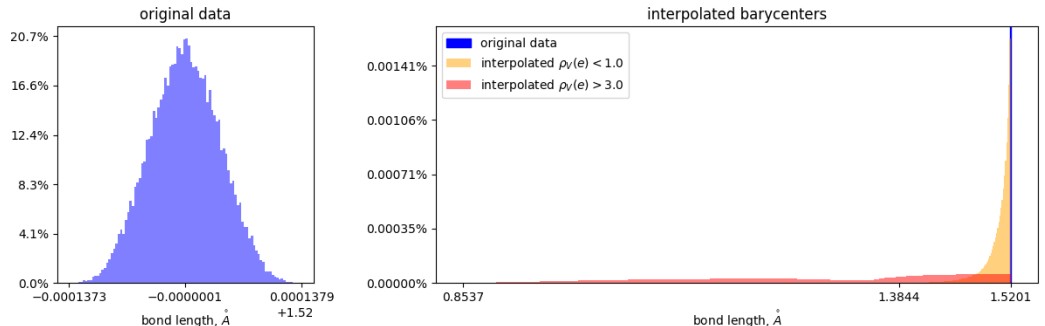

Figure 4: Distributions of bond lengths for conformations of cyclo-octane. Left: Bond lengths for the original data set, distributed tightly around 1.52 Å, standard deviation of $4.09 \times 10^{-5}$ Å. Right: Bond lengths for all barycenters of edges $e$ with $2\rho_V(e) < 1.0$, as would be allowed by the model $K_2$ described in Figure 3. The bond lengths vary from 1.3844 to 1.5201 Å with a mean of 1.5059 Å. Also, bond lengths for all barycenters of edges $e$ with $2\rho_V(e) > 3.0$, which are all allowed by the model $K_1$. The bond lengths vary from 0.8537 to 1.5201 Å with a mean of 1.2876 Å. The extremely narrow blue lines in the Right plot overlays the distribution from the original data set on the Left.

we penalize Wasserstein distance to $\text{PDiag}(R)$. For dots $(x, y) \in \text{PDiag}(L)$ with $x \leq \lambda_{\text{lo}} < \lambda_{\text{hi}} \leq y$, we penalize by the quantity $\|x - x_0\| \exp(-y)$, where $(x_0, y_0)$ is the best-match dot from $\text{PDiag}(R)$. This loss function should force $L$ to express long-lived topological features similar to those of $\text{PDiag}(L)$, while minimizing the error introduced at scales below $\lambda_{\text{lo}}$. As of this June 2023 publication, our code does not implement back-propagation to improve homological matching of models, but it is planned as upcoming work.

## 8   Example: Cyclo-Octane

The conformation space of cyclo-octane [18] is well-known to have novel topological structure. From physical principles, Martin et al. [22] show that real data sampled from the conformation space $X$ can be reduced to lie in $V = \mathbb{R}^{24}$, and furthermore $X$ must lie near a 2-dimensional stratified space consisting of a sphere and a Klein bottle. As in Section 1, we suggest that a machine-learning model $k : V \to \{0, 1\}$ trained to recognize cyclo-octane should not be considered "good" or trustworthy unless $K = \{x : k(x) = (1)\}$ also takes this geometric form at the appropriate scale. In this section, we demonstrate how topological parallax can support or reject the hypothesis that the geometry of a model $K$ matches the geometry of $X$.

Figure 3 visualizes the dataset $X \subset \mathbb{R}^{24}$ using a 3D Isomap projection.[3] Following the workflow from Section 5, we compute the 0-, 1-, and 2-dimensional persistence diagrams of $X \subset \mathbb{R}^{24}$ using gudhi [34], and we observe that there are highly persistent features—one 1-cycle and two 2-cycles.[4] The insight from [22] provides the meaning of these cycles, but the precise structure of the "data manifold" is typically not known *a priori* in examples. What we know in any case is that we want any potential $K$ to respect these cycles, because the geometry of $K$ should match the geometry of $X$, whatever it might be.

To understand the validity of a generated conformation, we compute its bond lengths–the distances between adjacent carbon atoms in the conformation. Bond length is an important physical property, together with bond angle, torsion angle, and energy [18]. Given the rigid geometry assumptions of the cyclo-octane data [22], we expect individual conformations generated by trained models to have similar bond lengths, and therefore the distribution of bond lengths from valid conformations should be similar to that of generated conformations. See Figure 4 to compare the bond lengths of conformations from the dataset $X$ versus those at barycenters for short edges and long edges in the Rips complex $R$ of $X$. Notably, interpolation across longer edges leads to invalid conformer geometries with too short of bond lengths and thus too sharp of bond angles for realistic molecules.

Suppose someone trains a standard neural network $k_1$ to recognize this data. For this demonstration, we used a 3-layer fully connected network with a ReLU and a SoftMax, implemented in PyTorch. The network was trained to near-perfect accuracy within a few minutes on a two class problem of real data versus a nearby background. (Hyperparameters and training details are provided in the Supplementary Material.) Thus, the model $k_1$ represents a common starting point that any data analyst might find encouraging. We apply Algorithm 7.2 to estimate which edges in $R$ are accepted by $k_1$, and discover $2\lambda_{\mathrm{lo},X}(K_1) = 3.45$, which is the longest edge available. So, the Rips complex cannot distinguish $K_1$ from the convex hull; the model does *not* reflect the geometry of $X$. This is particularly unfortunate in this case, because the model $k_1$ might be used to generate many new conformations (any interpolation between two valid conformations), the vast majority of which will not be valid conformations. An alternative model $k_2$ is offered, which is built from many local charts (details in Supplementary Material). The new model has $2\lambda_{\mathrm{lo},X}(K_2) = 1.0$. Moreover, the most persistent cycles in $X$ have infinite death as measured by the Rips-like path $L_t = P_{t,0}$, so $X$ and $K_2$ are homologically matched with $2\lambda_{\mathrm{hi}} \approx 1.25$. Therefore, we can claim that the geometry of $K$ matches the geometry of $X$ at these scales.

## Limitations

The parallax complex and associated objects are well-defined only for datasets and models that satisfy Definition 1.1. The algorithms in Section 7 assume that $V = \mathbb{R}^n$ with the Euclidean metric, but could be adapted for other geodesic spaces. Computation of the Rips complex and its persistence diagram scale favorably with the intrinsic dimension of the dataset $X$; however, the sampling methods discussed in Section 7 scale with the dimension of $V$, which might invoke the Curse of Dimensionality. A deeper question is whether real-life datasets $X$ of applied interest actually have enough sample density to exhibit a multiscale metric geometry, to which $K$ can be compared. This question suggests a followup study to verify whether datasets and models with demonstrated real-world efficacy actually have computable and comparable geometries; that is, future work should use parallax to assess the profound epistemological question of whether metric geometry is a valid metaphor for understanding deep learning in real-life applications.

## Acknowledgments and Disclosure of Funding

Work by all authors was partially supported by the DARPA AIE Geometries of Learning Program under contract HR00112290076, and by the National Institute of Aerospace (NIA) under sub-award C21-202066-GDA. We are very grateful to Bruce Draper of DARPA and Alywn Goodloe of NASA for their technical guidance during these efforts. We are also very grateful to Matt Dwyer, Rory McDaniel, Tom Fletcher, Yinzhu Jin, Jay Hineman, and Joey Tatro for technical discussions.

---

[3]Note that the pinch in the middle is an artifact of the projection and does not represent a singularity in the actual dataset.

[4]Gudhi uses diameter, not radius, as the filtration parameter.

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

# A  Supplementary Material: Formal Proofs

*Proof of Corollary 2.5.* The hypothesis $[Y] \mapsto [0]$ implies that for some $\alpha_0 \leq \alpha_2$ and $\varepsilon_0 \leq \varepsilon_2$, there exist chains $Y \in P_{\alpha_1,\varepsilon_1}$ and $Z \in P_{\alpha_2,\varepsilon_2}$ and $W \in P_{\alpha_0,\varepsilon_0}$ such that $\partial Z = Y - W$ in $P_{\alpha_2,\varepsilon_2}$. By Lemma 2.4, these chain can be included to their respective levels in $R$, giving $Y \in R_{\alpha_1}$, $Z \in R_{\alpha_2}$, and $W \in R_{\alpha_0}$, still satisfying $\partial Z = Y - W$ in $R_{\alpha_2}$. Hence, applying $\iota_*$, either $\iota_*([Y])$ is trivial in $HR_{\alpha_1}$, or the transition map $HR_{\alpha_1} \to HR_{\alpha_2}$ takes $\iota_*([Y]) \mapsto [0]$. $\square$

*Proof of Lemma 2.6.* If $\varepsilon \geq \alpha$, then the second condition in Definition 2.2 becomes $\rho_K(Y) - \rho_V(Y) \leq \alpha$, which is a trivial condition for $0 \leq \rho_V(Y) \leq \rho_K(Y) \leq \alpha$. Thus, the sets $P_{\alpha,\varepsilon}$ are identical for all $\varepsilon \geq \alpha$. $\square$

*Proof of Lemma 3.4.* Suppose an edge $e = (x_i, x_j)$ lies in $P_{\alpha,\varepsilon}$, meaning that $e$ is represented by a $V$-geodesic of length $2\rho_V(e)$ satisfying $\rho_V(e) \leq \alpha$ and a $K$-geodesic of length $2\rho_K(e)$ satisfying $\rho_V(e) \leq \rho_K(e) \leq \min\{\alpha, \rho_V(e) + \varepsilon\}$. By Lemma 3.3, the corresponding edge $e' = f_\sharp(e) = (f(x_i), f(x_j))$ satisfies $\rho_V(e) - \kappa \leq \rho_V(e') \leq \rho_V(e) + \kappa$. Moreover, by the data perturbation lemma on $K$, we know $e'$ is represented by a $K$-geodesic with length $2\rho_K(e')$ satisfying $\rho_K(e') \leq \rho_K(e) + \kappa$. Combining these overall, we have the parallax bounds

$$\rho_V(e') \leq \rho_K(e') \leq \min\{\alpha, \rho_V(e) + \varepsilon\} + \kappa$$

which implies both $\rho_K(e') \leq \rho_V(e') + \varepsilon + 2\kappa$ and $\rho_K(e') \leq \alpha + \kappa$. $\square$

*Proof of Lemma 4.6.* Fix $\lambda' = \frac{1}{2}\lambda_{\mathrm{ball},X}(K) - \kappa$. We aim to show that $X'$ and $K$ are $\lambda'$-LSM. Fix any edge $e' = (x', y')$ in $R'$ satisfying $\rho_V(e') \leq \lambda'$. (The set of such edges might be empty, which is allowed by Definition 4.1.) For any such $e'$, there is a corresponding $e = (x, y) \in R$ with $f(x) = x'$, $f(y) = y'$, and $f_\sharp(e) = e'$. The triangle inequality provides $\|y - x\| \leq \|y - y'\| + \|y' - x'\| + \|x' - x\| = 2\lambda' + 2\kappa \leq \lambda_{\mathrm{ball},X}(K)$. We are assuming $V$ is Euclidean, so all balls in $V$ are convex. Since $x', y, y'$ are all within the ball of radius $\lambda_{\mathrm{ball},X}(K)$ around $x$, all their pairwise geodesics are included in $K$. Hence, $e' \in P'_{\lambda',0}$. $\square$

*Proof of Lemma 5.2.* The relationship $c \leq d$ follows from Lemma 2.5. The relationship $d \leq e$ follows from Lemma 2.3 and the observations that $P_{\alpha,\alpha} = P_{\alpha,\infty} = R(X, K)$. Suppose $L$ is parameterized as $L_\alpha = P_{\alpha,\varepsilon(\alpha)}$. Suppose that $c < e$, so either $c < d$ or $d < e$.

If $c < d$, then we can conclude that the edge that killed $[Y]$ in $HR(X, V)$ is not present in $HR(X, K)$, so that edge intersected $K^c$, which is an open set. Hence, that killing edge intersects a void in the sense of Defn 1.2. Let $B_r$ be an open ball in $K^c$ centered on some point on the $V$-geodesic of length $2c$. If $B_r = K^c$, then the shortest $K$-geodesic replaces the diameter $2r$ with the half-circumference $\pi r$. Therefore, $\pi r - 2r \leq 2d - 2c$.

If $d < e$, then $[Y] \mapsto [0]$ via $HP_{b,b} \to HP_{d,d}$ and via $HP_{b,b} \to HP_{e,\varepsilon(e)}$, but not for any $HP_{b,b} \to HP_{\alpha,\varepsilon(\alpha)}$ with $\alpha < e$. Therefore, there is a $K$-geodesic of length $2d$ and a different $K$-geodesic of length $2e$, either of which could kill $[Y]$. The former edge is not allowed in $L$ because $0 \leq \varepsilon(d) < d - c$. Hence, $c < d$, returning us to the first case. $\square$

*Proof of Lemma 5.3.* Let $L$ be a Rips-like path given by Definition 5.1. Because $X$ and $K$ are $\lambda$-locally simplicially matched, the filtrations levels of $R$ and $L$ are identical up to $\lambda$. Therefore, there is a bijection between the persistence diagrams of $R$ and $L$ for all dots born by $\lambda$.

Suppose that $(b, d)$ is a dot in the persistence diagram of $L$ with $b \leq \lambda < \omega < d$. This dot represents a class $[Y]$ born in $HL_b$ that dies in $HL_d$. The class $[Y] \in HR_b$ is born at $b$ must die at some value $c$, so $[Y] \in \ker HR_{b \to c}$. It cannot be that $c \leq \delta$, because that would imply $[Y] \in \ker HL_{b \to \omega}$, contradicting $\omega < d$. Therefore, $\delta < c$. $\square$

*Proof of Theorem 5.4.* Let $f_* : HP \to HP'$ denote the map induced by $f$ and similarly, let $f_*^{-1} : HR' \to HR$ denote the map induced by $f^{-1}$. Consider the following commutative diagram in which

the unlabeled morphisms are transition maps of the respective persistence modules.

$$
\begin{array}{ccc}
HR'_{\delta-\kappa} & \xrightarrow{\ f_*^{-1}\ } & HR_\delta \\
\uparrow & & \uparrow \\
N' \xrightarrow{\ \subset\ } HR'_{\lambda-\kappa} \xrightarrow{\ f_*^{-1}\ } HR_\lambda = HP_{\lambda,\epsilon_1} & \longrightarrow & HP_{\omega,\epsilon_2} \\
\Big\downarrow = & \Big\downarrow f_* & \Big\downarrow f_* \\
HP'_{\lambda-\kappa,\epsilon_1+2\kappa} \longrightarrow HP'_{\lambda+\kappa,\epsilon_1+2\kappa} & \longrightarrow & HP'_{\omega+\kappa,\epsilon_2+2\kappa}
\end{array}
$$

Note that the bi-degrees in the diagram are determined by Lemma 3.3 and Corollary 3.5. Set $N' = \ker HR'_{\lambda-\kappa\to\delta-\kappa} \subset HR'_{\lambda-\kappa}$. Commutativity of the top left square implies that $f_*^{-1}(N') \subset HR_\lambda$ maps vertically to 0 in $HR_\delta$ and hence, $f_*^{-1}(N')$ lies in $\ker HR_{\lambda\to\delta}$. The homologically matched assumption on $(X,K)$ implies $f_*^{-1}(N') \subset \ker HP_{(\lambda,\epsilon_1)\to(\omega,\epsilon_2)}$. By commutativity of the bottom right square, $f_*(f_*^{-1}(N'))$ maps horizontally to $0 \in HP'_{\omega+\kappa,\epsilon_2+2\kappa}$. Finally, the locally simplicially matched assumption means we can instead think of $N'$ as a subset of $HP'_{\lambda-\kappa,\epsilon_1+2\kappa}$ which maps to 0 in $HP'_{\omega+\kappa,\epsilon_2+2\kappa}$ by commutativity. Thus,

$$
N' = \ker HR'_{\lambda-\kappa\to\delta-\kappa} \subset \ker HR'_{(\lambda-\kappa,\epsilon_1+2\kappa)\to(\omega+\kappa,\epsilon_2+2\kappa)}\,,
$$

completing the proof. $\qquad\square$

## B  Supplementary Material: Code

The code supporting this article is an initial proof-of-concept. It is available publicly at



https://gitlab.com/geomdata/topological-parallax



It is designed as a simple Python package, following community best-practices for tooling and layout. Documentation is provided there. We recommend that the reader go to the repository for issue tracking, improvements, bug fixes, testing pipelines, etc.

Because this package relies on GUDHI The GUDHI Project [34], the filtration values are by diameter (not radius). This is important to note, because the theory in the paper is written using radius (not diameter) as the filtration value.

Topological parallax has the same computational complexity as the computation of Rips complexes and their persistence diagrams—albeit with a larger constant. This is because parallax merely inserts a model-evaluation step upon the examination of each edge. The constant therefore is $tN^2$ for $N$ points and a model that takes time $t$ to evaluate. There are very interesting dimension- and structure-dependent estimates for the real-life/expected timing of Rips computations, such as Bauer et al. [4]. Distributed persistence can parallelize this process, as in Solomon et al. [31]. See https://gitlab.com/geomdata/dispers.

- Figure 1 was produced with the Jupyter notebook `notebooks/Winky Example.ipynb`. Compare and contrast the Python scripts `runners/run_winky_nn.py` versus `runners/run_winky_tree.py`

- Figure 3 was produced with the Python script `runners/run_cyclooctane_balls.py`. Compare and contrast that script with `runners/run_cyclooctane_nnfar.py` and `run_cyclooctane_nnhull.py`. These examples may take 100-500 GB of RAM to compute the persistence diagrams as currently implemented.

- Figure 4 was produced with the Python script `runners/bond_lengths.py`

# C  Supplementary Material: Utah Jar Example

This example demonstrates a situation where a classifier appears to be 100% correct on a semantically meaningful dataset, but the resulting model is too strict for interpolation within a data class. Topological parallax detects this phenomenon, as discussed in Section 6.

We applied parallax to an imagery dataset inspired by the "Utah teapot." The dataset consists of 14000 images—4000 images of a rotated teapot and 10000 images of a rotated teapot with no spout or handle (referred to as the "Utah jar"). See Figure 5a. All images were rendered with PoV-Ray.

We constructed a bespoke classifier on this dataset of images which accepts any image within a distance of 1.0 to the set of jar images using the Euclidean metric on flattened images. Our classifier rejects all other inputs. We applied parallax to this model and found very small values for both $\lambda_{\text{lo}}$ and $\lambda_{\text{sup}}$, with $\lambda_{\text{lo}} \sim \lambda_{\text{sup}} < 2$. See Figure 5b.

Given the hard cut-off distance of 1.0 in the classifier and these $\lambda$ values, we expect the model to prohibit reasonable interpolations of images. Indeed, we found numerous interpolations rejected by the model which seem valid to the human eye. See Figure 5c.

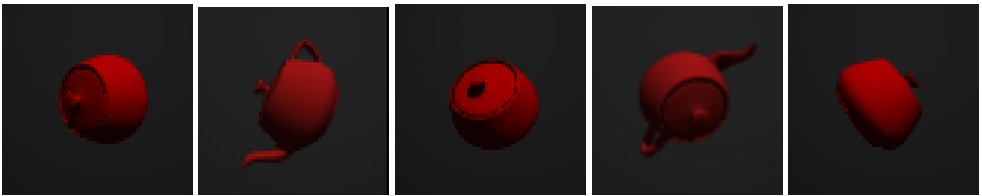

(a) Examples from the modified Utah Teapot dataset.

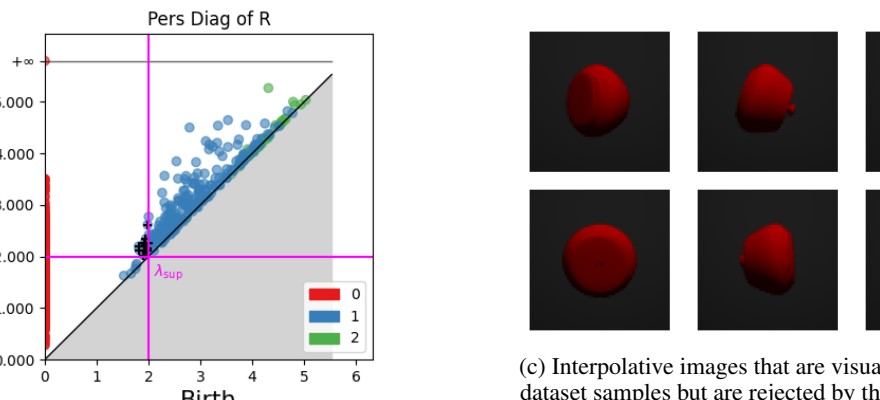

(b) The persistence diagram of the dataset marked by parallax.

(c) Interpolative images that are visually similar to true dataset samples but are rejected by the overly sensitive model.

Figure 5: Using parallax to detect a dataset-model geometry mismatch. (a) The images consist of the Utah teapot (rendered with PoV-Ray) with spouts and handles removed for simplification. (b) We construct a simple classifier which accepts any images within a Euclidean distance of 1.0 of the (flattened) image and rejects all others, and apply parallax to it. (c) Six interpolated images are shown, all of which look very similar to the original dataset but are rejected by the model (corresponding to the X marks in (b)).

# D  Supplementary Material: Understanding the Bi-Complex

The parallax complex $P(X, K, V)$ is bi-filtered by the parameters $\alpha$ (the length of a geodesic) and $\varepsilon$ (the distortion of geodesic length between $K$ and $V$). Figure 6 provides some visualizations that may help the reader interpret traditional barcodes of Rips-like paths, and how these bi-filtration parameters may be estimated. These parameters are related to the size of voids in the model, as seen in Lemma 5.2.

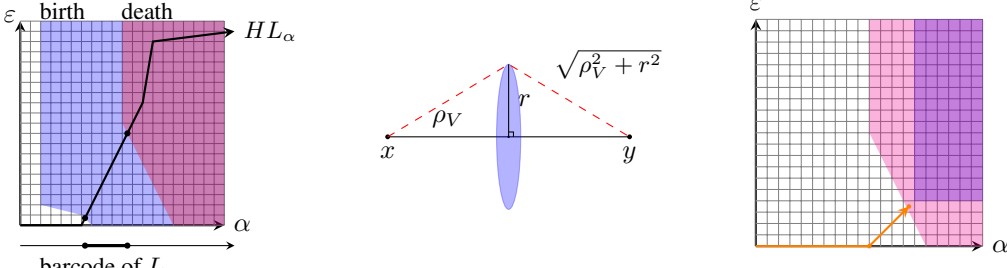

Figure 6: Left: Births and deaths are bi-filtered in $HP$, and are observed by barcodes on Rips-like paths. Cor 3.5 means this picture is stable within $\pm(\kappa, 2\kappa)$. Center: An edge $(x, y) \in P_{\alpha,\varepsilon}$ can be estimated by sampling tubular neighborhoods. Right: Algorithm 7.5 underestimates $\alpha, \varepsilon$.

# E Supplementary Material: Table of Symbols

| Notation | Plain Meaning | First Appearance | Term |
|---|---|---|---|
| $V$ | A geodesic space, such as $\mathbb{R}^n$ | p.2 | Ambient Space |
| $X$ | A finite set in $V$ | p.2 | Dataset |
| $k$ | A perception function on $V$ | p.2 | Model (as function) |
| $K$ | Support set of $k$ | p.2 | Model (as set) |
| $\mathcal{M}(X)$ | Models compatible with dataset $X$ | p.2 | |
| $\mathcal{M}^*(K)$ | Datasets compatible with model $K$ | p.2 | |
| $K^\circ$ | Interior of set $K$ in topological space $V$ | p.2 | |
| $\overline{K}$ | Closure of set $K$ in topological $V$ | p.2 | |
| $K^c$ | Complement of set $K$ in $V$ | p.2 | |
| $\Omega$ | Bounded open set in $K^c$ | p.2 | Void |
| $R(X, K)$ | Rips complex of $X$ in geodesic space $K$ | p.4 | |
| $\alpha$ | a filtration level or radius | p.4 | |
| $B_\alpha(x)$ | Geodesic ball of radius $\alpha$ about $x$ | p.4 | |
| $Y$ | Chain in a Rips complex | p.4 | |
| $\rho_K(Y)$ | Minimal filtration radius for $Y$ in $R(X, K)$ | p.4 | |
| $\varepsilon$ | Gap in filtration radius between $V$ and $K$ | p.4 | |
| $P$ | Parallax bi-complex for $X, K, V$ | p.4 | Parallax |
| $HP$ | Homology of $P$ | p.4 | |
| $L$ | A 1-parameter path through $P$ | p.4 | Rips-like Path |
| $HL$ | Homology of $L$ | p.4 | |
| $\overset{\kappa}{\approx}$ | Pointwise perturbation of $X$ in $V$ | p.5 | Perturbation |
| $\overset{\kappa}{\approx}_K$ | Pointwise perturbation of $X$ in $K$ | p.5 | $K$-Perturbation |
| $f_\sharp$ | Induced map on a simplicial complex | p.5 | |
| $f_*$ | Push-forward map on homology | p.5 | |
| $\lambda_\bullet$ | A meaningful filtration value. See subscript. | p.6 | |

