# OpenReview forum: "Topological Parallax: A Geometric Specification for Deep Perception Models"
_NeurIPS.cc/2023/Conference — NeurIPS 2023 spotlight_

### Official Review · Reviewer_Hzkd · 2023-07-04

**Soundness:** 3 good
**Presentation:** 3 good
**Contribution:** 3 good
**Rating:** 7
**Confidence:** 3

**Summary:**

The paper presents a framework for analyzing the validity of deep classification models by comparing its multi-scale geometric features with that of the training dataset. The comparison is conducted through tools of topological data analysis, mainly using persistence diagrams to detect and characterize the nontrivial topological features (connected components, 1-cycles, 2-cycles, etc.) as they are revealed by changing the radius of data points to construct Ribs complex. The main challenge is to conduct such analysis on the implicit space of the deep classification model; the paper addresses this challenge by devising distance estimate algorithms for the implicit space, and comparing it with the ambient Euclidean distance to decide the matching of geometric features. The paper shows an application of the analysis on a single dataset and accompanying simple classification models.

**Strengths:**

The paper asks the important question of how to decide if a deep perception model faithfully captures the true characteristics of a dataset. It proposes to check if the learned implicit space of the perception model has geometric features that closely match the dataset distribution.

The paper develops a theoretical framework to characterize the geometric matching of model and data. The important notion is to construct a parallel complex, that is obtained by the co-filtration of Rips complex by metrics of both the data set and the learned model. Based on this construction, the comparison of model and data becomes homological computations on the series of complexes, which draws on topological data analysis extensively.

The paper discusses applications through a concrete example, and of the possibility of using the assessment as objective function for training more accurate networks.

**Weaknesses:**

The paper is theoretical in nature. To ease understanding, it is better to provide tables to summarize the notations introduced and use more figures to motivate and illustrate the ideas. I find this can be helpful particularly for the perturbation lemmas and local simplicial matching. Drawing more connections to TDA can be helpful too, to motivate the constructions introduced in this paper.

After developing the theoretical framework, it is desirable to immediately position Sec.6 in context, maybe through the example of Sec.8 or other cases.

More application examples can be used. The cyclo-octane dataset has a very unintuitive structure. If more intuitive datasets and perception tasks can be tested, the usefulness of this analysis can be more convincing. Such intuitive datasets can come from computer vision data, e.g. MNIST, CIFAR.

Deep learning models are witnessing a shift from perception to generative modeling, achieved mostly through minor variations in objective function and output format. How do the motivation and framework proposed by this paper apply to generative models? This paper does not include any such discussion.

**Questions:**

The major questions have been posed in the above discussions about weaknesses.

Minor questions:

Definition 1.1, $K^\circ, \overline{K^\circ}$ should be defined clearly. A summary of important notations could be given in the text.

Line54, the order of $M$ and $M^*$ seems to be reversed?


**Limitations:**

The authors have discussed limitations extensively.

---

> ### Author Rebuttal · Authors · 2023-08-09
>
> ### Regarding readability
>
> We thank the reviewer for their comments regarding readability and
> understandability. We will include a table of notation in the Supplemental Material to help
> the reader understand our mathematical notation. We are also planning on
> replacing Figure 2 with Figure 1 in the attached PDF to provide a high-level
> understanding of parallax, as well as the importance of matching the geometry
> between model and data. We will also expand our references to include standard
> TDA textbooks, such as Edelsbrunner and Harer's computational topology
> textbook, as well as Hatcher's algebraic topology text.
> We will also add a preview of section 6 and 8 to the introduction to help orient the reader.
>
> ### Regarding additional examples
>
> We agree that additional applications and examples (particularly in imaging data) are desirable to advance the field.
> Please see the general rebuttal for the description of another imaging dataset example.
> We are preparing a separate manuscript that surveys imaging data and popular vision networks.
> The rich field of convolutional neural networks deserves special attention due to the high extrinsic dimension and counter-intuitive metric geometry.
> In preparing this submission,
> we found that discussion of the often counter-intuitive metrics took too much attention away from the important discussion of topological interpretation and stability analysis.
> In this submission, we intentionally restricted the scope to an introduction to the core definitions and algorithms, and we demonstrated their meaning on examples that did not rely on convolutional layers,
> but we look forward to presenting followup work on imaging data in the near future.
>
> ### Regarding application to generative modeling
>
> We agree that generative modeling is increasingly important, especially in the context of adversarial robustness and safety.
> We are preparing a separate manuscript on that topic, but the mathematical formulation of generative models is more subtle than perception models.
> In this submission, we restricted the scope to perception models to keep the (already heavy)
> mathematical formulation manageable within the space available.
>
> ### Regarding minor comments
>
> We thank the reviewer for catching these mistakes. We will be sure to define all of the notation used in Definition 1.1
> and we will add a table summary of notation used to the Supplemental Material.
> Indeed, the order of \\(\\mathcal{M}\\) and \\(\\mathcal{M}^*\\) in Line 54 is reversed and should be corrected.

---

> > ### Comment · Reviewer_Hzkd · 2023-08-15
> >
> > Thanks for the response. I remain positive about the submission.

---

### Official Review · Reviewer_xrbA · 2023-07-06

**Soundness:** 2 fair
**Presentation:** 1 poor
**Contribution:** 2 fair
**Rating:** 5
**Confidence:** 1

**Summary:**

The authors suggest that a model is good if its geometry matches the geometry of data, and introduce a persistent-homology based method to evaluate this similarity.

**Strengths:**

There is a number of theoretical results that seem to support the soundness of the proposed approach.

**Weaknesses:**

(W1) The paper is extremely hard to read, which significantly limits its impact. For example, you mention that you rely on bi-filtered persistence module, but it is still unclear to me what the two filtrations are (what is reflected by alpha and epsilon), and why each of them is useful. As another example, rho_K(Y) is a crucial notion for your work, but you never name or describe it, let alone motivate it or provide intuitive explanation; also, this requires an explicit Definition. As a final example, Figure 2 remains a mystery.

All definitions and lemmas could be named, but more importantly, they could be first motivated and then interpreted. The latter could be done in the beginning of a section, by summarizing all the results and their relevance in a paragraph or two, or throughout the section in a story-like manner. The same holds for the whole text, e.g., when you describe your procedure, you could describe the concepts in words (even if you did this before, it is good to remind the reader, especially since the paper is extremely notation-heavy):  (2) for a model [nice] K ∈ M(X), compute [what is this] λlo,X(K) and [what is this] λhi,X(K)”, or “In this section, we provide algorithms to estimate [what is this] P_{alpha, epsilon}”. A well-organized notation table might also be helpful.

For this reason, a lot of my feedback is an educated guess, and I am definitely open to substantially changing my score for this paper if it is presented more clearly, so that I can actually assess the soundness and contributions.

(W2) Experiments consider only one data set.


**Questions:**

(Q1) Can you motivate the name Parallax?

(Q2) You write: “We suggest that a model K is ‘good’ if the geometry of K matches the geometry of X.” How is this related to “If a certain architecture is incapable of expressing a decision region that is equivalent in topology to training data, then there is no hope of it ever generalizing to the true data” in [1]? More generally, articles [1]-[3] also consider persistent homology of the data and model, can you comment if and how they are related to your approach?

(Q3) Can you elaborate on the relationship between your work and the related articles you mention in Section 1.3, in particular with [17] and [28]?

(Q4) Where is the proof of Lemma 2.6?

(Q5) “In Section 4, we introduced ‘local simplicial matching’ as a way to compare small-scale geometry. In this section, we introduce ‘homological matching’ as way to compare large-scale geometry.” Should this not be mentioned earlier in the paper? Is it possible to provide an illustration?

(Q6) “We apply Algorithm 7.2 to estimate which edges in R are accepted by k1, and discover 2λlo,X(K1) = 3.45, which is the longest edge available. So, the Rips complex cannot distinguish K1 from the convex hull; the model does not reflect the geometry of X.” How do you make this conclusion, can you elaborate? This is related to weakness (W1).


Other minor comments:

- Line 28: should be applied -> could be applied?
- Figure 1: It would be better if the axes ranges are also provided in the left plot, to make a connection with its PD.
- Line 168: Def’n -> Definition
- Explicitly mention that the code is made publicly available.
- Is Limitations Section 9?


[1] Guss, William H., and Ruslan Salakhutdinov. "On characterizing the capacity of neural networks using algebraic topology." arXiv preprint arXiv:1802.04443 (2018).

[2] Ramamurthy, Karthikeyan Natesan, Kush Varshney, and Krishnan Mody. "Topological data analysis of decision boundaries with application to model selection." International Conference on Machine Learning. PMLR, 2019.

[3] Khrulkov, Valentin, and Ivan Oseledets. "Geometry score: A method for comparing generative adversarial networks." International Conference on Machine Learning. PMLR, 2018.


**Limitations:**

Limitations are discussed in the final section.

---

> ### Author Rebuttal · Authors · 2023-08-09
>
> ### Re: the bi-filtration
>
> The bifiltration is the most technically challenging concept in this project, but necessary for the stability results (Lemma 3.4 and Theorem 5.4).
>
> The first parameter \\(\\alpha\\) is the filtration radius (that is, half the length) of \\(K\\) geodesic.
> The second parameter \\(\\varepsilon\\) is the difference between the length of that \\(K\\) geodesic and the corresponding \\(V\\) geodesic.
> This meaning is embedded in Definition 2.2.
>
> Consider two antipodal points on a unit circle, where the model is a tight annulus matching that circle.
> In the ambient \\(V\\), the Rips edge between those two antipodal points would have filtration radius 1.
> In the annular model \\(K\\), the Rips edge between those two points would have filtration radius \\(\\approx \\pi/2\\), because the geodesic has to go around the circle, so \\(\\varepsilon \\approx \\pi/2-1\\\) in this case.  \\(\\varepsilon\\) keeps track of the local distortion of length between \\(V\\) and \\(K\\). This can be used to estimate the size of voids and perhaps be used to make estimates of the curvature of \\(\\partial K\\), for example. We would be happy to elaborate upon the meaning of the \\(\\varepsilon\\) parameter briefly, near Definition 2.2.
> We could also discuss the use of \\(\\varepsilon\\) to estimate the size of voids in the Supplemental Material.
>
> ### Re: \\(\\rho_K(Y)\\)
>
> The definition of \\(\\rho_K(Y)\\) is in lines 105--107, where the Rips complex is defined.
> For a particular edge \\(e=(x_0, x_1)\\),
> \\(\\rho_K(e)\\) is half the geodesic distance between \\(x_0\\) and \\(x_1\\) in \\(K\\).
> We agree that this jargon can be difficult for readers outside our speciality.  We will help orient the reader by providing this brief clarification along with a reference.
>
> ### Re: Figure 2
>
> Figure 2 is intended to show the structure of the bi-filtration, and how births
> and deaths of the bifiltration can be detected using TDA. Figure 2 is not crucial, and we will replace it with
> Figure 1 of the included PDF.
>
> ### Re: naming of lemmas
>
> We agree that this is a good strategy, and we named the definitions, theorems, etc. which we see as the most crucial statements.
> A minority of statements
> (2.1, 2.4, 2.6, 4.2, 4.3, 4.4, 4.5, 4.6, 5.2, 5.4, 7.1) were unnamed because they provide intermediate technical bounds needed elsewhere.
> We are open to additional suggestions for which statements deserve specific names.
>
> ### Re: motivation
>
> Sections 1.1 and 1.2 were intended
> to give this sort of story-like motivation and summary, but
> it is clear from reviewer comments that they could be improved. We
> will rephrase section 1.2 to provide a clear
> roadmap of the major definitions and lemmas, and add a table of notation to
> the Supplemental Material to help orient a reader. We will replace Figure 2
> with Figure 1 of the attached PDF to help elucidate the main ideas behind parallax
> and the importance of understanding the geometry of models.
>
> ### Re: (W2)
>
> We agree that the paper would be improved with further exploration of parallax on other datasets,
> in particular on imagery datasets. We are planning a suite of experiments in which
> we will apply parallax to a broad set of commonly used imaging data, but could
> not include it here due to page constraints. The present work is meant to
> introduce the main definitions and provide justification for the theorems and ideas of the paper.
>
> ### Re: (Q1)
> Yes! Parallax was named by analogy to the method in astronomy. There is an
> inaccessible object that cannot be measured directly (in this case, the
> geometry of the model \\(K\\)), so we must infer its location by comparing
> multiple observations from the available vantage points (in this case, the
> points in the dataset \\(X\\)).  We will add a sentence to this effect
> to the introduction.
>
> ### Re: (Q2)
>
> We thank the reviewer for reminding us of this paper and we will
> reference it. The key difference between this work and ours is that our theory quanitifies the
> ability of a specific model, independent of parameters or network architecture, to match
> the shape of a dataset in a specified way. The Guss et al. paper discusses the ability of
> any model produced from a specific network architecture to perform well on datasets of a specific
> shape.
>
> ### Re: (Q3)
>
> Both of these references track and study the topology of a dataset as it evolves
> through the layers of a neural network. They make empirical claims about what
> tends to happen as training accuracy increases, but do not provide
> geometric specifications about what should happen if one is to trust the model,
> and these specifications are sorely needed because, as has been shown in the literature,
> metrics such as perfect training accuracy are not sufficient.
>
> ### Re: (Q4)
>
> The proof is as follows, and should be added to the Supplemental Material or immediately after Lemma 2.6.
> If \\(\\varepsilon \\geq \\alpha \\), then the second conditions in Definition 2.2 becomes \\(\\rho_K(Y) - \\rho_V(Y) \\leq \\alpha\\),
> which is a trivial condition for \\(0 \\leq \\rho_V(Y) \\leq \\rho_K(Y) \\leq \\alpha\\).
> Thus, the sets \\(P_{\\alpha,\\varepsilon}\\) are identical for all \\(\\varepsilon \\geq \\alpha\\).
>
> ### Re: (Q5)
>
> Yes, we should have foreshadowed these terms in Section 1.
> Figure 1 of the attached PDF could be added in Section 1 or in the Supplemental Material to clarify the interpretation.
>
> ### Re: (Q6)
>
> Yes. Since \\(\\lambda_{lo}\\) equals the filtration level of the longest
> edge, all edges are included in the parallax complex. For every edge in the
> Rips complex of the original dataset, the geodesic representing that edge is
> contained in the model. Thus, the Rips complexes \\(R(X,V)\\) and
> \\(R(X,K)\\) are identical.  This does not imply that \\(K=V\\), but any
> differences are undetected by the pairwise geodesics between the points of
> \\(X\\).
>
> ### Re: Minor Comments
>
> We thank the reviewer for these wording changes and figure formatting comments, and
> we will be happy to make them.

---

> > ### Comment · Reviewer_xrbA · 2023-08-14
> >
> > I appreciate the detailed response and the additional clarifications (please make sure to include them in the paper, e.g. comments on the bi-filtration, \rho, the name parallax). As indicated earlier, I raised my rating.
> >
> > However, I still worry whether the authors are fully aware of the readability issues of the paper, even though they were raised by multiple reviewers and acknowledged in the general comment. For example, you say that the definition of \rho is in lines 105 -- 107, but the point I was trying to make is that it should be named and included in a separate Definition environment, since it is a crucial concept for the paper (any two lines in the text are easy to miss, and additional references are not going to help much here). I am curious to see the promised notation table.

---

> > > ### Author Response · Authors · 2023-08-16
> > >
> > > We thank the reviewer for the additional feedback. We take the readability
> > > concerns seriously. We have reserved the definition field in latex for
> > > definitions novel to the paper. Perhaps additional bolding or highlighting
> > > would help draw the reader's attention to the first use of mathematical jargon.
> > >
> > > The first sentence of Section 2 could be improved with the following
> > > introductory comment:
> > >
> > > "In the geodesic space \\(V\\), \\(B_{\alpha}(x)\\)
> > > denotes the geodesic ball of radius \\(\alpha\\) around \\(x\\). For a formal
> > > edge \\(e=(x_0,x_1)\\) between points in \\(X\\), \\(\rho_V(e)\\) is the
> > > minimum radius for which \\(B_{\rho_V(e)}(x_0)\\) intersects
> > > \\(B_{\rho_V(e)}(x_1)\\). Thus, \\(2 \rho_V(B)\\) is the geodesic distance in
> > > \\(V\\) between \\(x_0\\) and \\(x_1\\). The Rips complex \\( R(X,V) \\) is the
> > > simplicial complex generated by these edges, as filtered by \\(
> > > \rho_V(e)\\)[13, Section III.1]. More generally, for any \\(K \in
> > > \mathcal{M}(X)\\) ..."
> > >
> > > Here is an example notation table that might help orient the reader.
> > >
> > > | Notation | Plain Meaning | First Appearance | Term |
> > > |----------|---------|------------------|-----|
> > > |\\(V\\) | A geodesic space, such as \\(\\mathbb{R}^n\\) | p.2| Ambient Space |
> > > |\\(X\\) | A finite set in \\(V\\) | p.2| Dataset ||
> > > |\\(k\\)| A perception function on \\(V\\) | p.2| Model (as function) |
> > > |\\(K\\)| Support set of \\(k\\) | p.2| Model (as set)
> > > |\\(\\mathcal{M}(X)\\)| Models compatible with dataset \\(X\\)| p.2|
> > > |\\(\mathcal{M}^*(K)\\)| Datasets compatible with model \\(K\\) | p.2|
> > > |\\(K^{\circ}\\)| Interior of set \\(K\\) in topological space \\(V\\)| p.2|
> > > |\\(\overline{K}\\)| Closure of set \\(K\\) in topological \\(V\\)| p.2|
> > > |\\(K^{c}\\)| Complement of set \\(K\\) in \\(V\\)| p.2|
> > > |\\(\Omega\\)| Bounded open set in \\(K^c\\) | p.2| Void |
> > > |\\(R(X,K)\\)| Rips complex of \\(X\\) in geodesic space \\(K\\)|p.4|
> > > |\\(\\alpha\\)|a filtration level or radius|p.4|
> > > |\\(B_{\alpha}(x)\\)|Geodesic ball of radius \\(\\alpha\\) about \\(x\\)|p.4|
> > > |\\(Y\\)| Chain (formal sum of simplices) in a Rips complex | p.4| |
> > > |\\(\rho_K(Y)\\)|Minimal filtration radius for \\(Y\\) in \\(R(X,K)\\) |p.4||
> > > |\\(\\varepsilon\\)|Difference of filtration radius between \\(V\\) and \\(K\\)|p.4|
> > > |\\(P\\)|Parallax bi-complex for \\(X, K, V\\)|p.4|Parallax|
> > > |\\(HP\\)|Homology of \\(P\\)|p.4|
> > > |\\(L\\)|A 1-parameter path through \\(P\\)|p.4|Rips-like Path|
> > > |\\(HL\\)|Homology of \\(L\\)|p.4|
> > > |\\(\\overset{\\kappa}{\\approx}\\)|Pointwise perturbation of \\(X\\) in \\(V\\)|p.5|Perturbation|
> > > |\\(\\overset{\\kappa}{\\approx}_K\\)|Pointwise perturbation of \\(X\\) in \\(K\\)|p.5|\\(K\\)-Perturbation
> > > |\\(f_{\sharp}\\)|Induced map on a simplicial complex |p.5|
> > > |\\(f_*\\)|Push-forward map on homology |p.5|
> > > |\\(\lambda_{.}\\)|A meaningful filtration value in \\(HP\\). E.g., "lo", "ball", "sup", "hi"|p.6|

---

### Official Review · Reviewer_3L38 · 2023-07-07

**Soundness:** 3 good
**Presentation:** 3 good
**Contribution:** 4 excellent
**Rating:** 8
**Confidence:** 3

**Summary:**

This paper introduces topological parallax, a theoretical framework for analyzing the similarity of multiscale geometric structures between datasets and models. It estimates the topological features in the model by examining the effect on the Rips complex of geodesic distortions using the reference dataset. It shows the stability of the proposed framework under dataset perturbations. It also provides a practical computational method on top of the theory.

**Strengths:**

- The paper provides a novel point of view. It claims that it is the first work to use TDA to express a desired geometric relationship between models and datasets, and to the best of my knowledge, I don't see any existing work studying this problem.
- This paper provides interesting insights into the concepts of "overfitting" and "generalization" of neural networks, which are important for the safety and robustness of AI.
- It proposes a theoretical framework with abundant derivations and proofs.
- It also introduces a practical computational method and demonstrates it on concrete data and network examples.

**Weaknesses:**

- Section 8 shows an interesting example of a data space with novel topological structures. But is this method also applicable to other real-world scenarios such as image recognition? What is the complexity of the computation w.r.t. dataset size and dimensionality?

**Questions:**

- The theoretical framework is built upon a binary classification problem. Is it potentially generalizable to more complex model outputs, e.g. multi-class classification (especially when different output channels are inter-dependent)? How would the geodesics be defined in such a case?

**Limitations:**

Limitations are well-discussed in the paper. As a theory work, I believe it won't have direct social impacts.

---

> ### Author Rebuttal · Authors · 2023-08-09
>
> ### Regarding real-world scenarios such as image recognition
>
> Yes!  See the general rebuttal for a discussion of further experimentation with imaging datasets.
>
> ### Regarding computational complexity
>
> The computation of parallax is the same "big-O" as the computation of Rips complexes and their persistence diagrams---albeit with a larger constant.
> This is because parallax merely inserts a model-evaluation step upon the examination of each edge.
> The constant therefore is \\(tN^2\\) for \\(N\\) points and a model that takes time \\(t\\) to evaluate.
> There are very interesting dimension- and structure-dependendent estimates for the real-life/expected timing of Rips computations,
> https://arxiv.org/abs/2211.09075
> and, we would be happy to include a brief discussion of these considerations in either Section 7 or the Supplemental Material.
>
> ### Regarding multi-class classification
>
> This is a very interesting question.
> The theory is presented in this submission for single-class perception problems,
> but as emphasized by the reviewer,
> many key applications will involve multi-class problems (such as overlapping families in multi-label imaging datasets).
> To handle these situations, we typically consider each label separately (against the others) or study in semantically meaningful collections of label.
>
> The overall theme in this case is that the geometry of each class and the combined geometry of relevant mixtures of classes should be respected by the respective models.
> One can manipulate the definition of the model oracle \\(k\\) to account for relative likelihoods of various labels, and then study the output of parallax as those relative likelihoods vary.
>
> We can include a brief discussion of this consideration in the Supplemental Material.
> A more detailed analysis of multiclass problems, especially in the context of imaging datasets, will occur in a forthcoming manuscript focussed on that topic.

---

> > ### Comment · Reviewer_3L38 · 2023-08-22
> >
> > Thanks for your reply! I think my questions are well-answered, so I'll keep my positive rating.

---

### Official Review · Reviewer_GC8k · 2023-07-07

**Soundness:** 3 good
**Presentation:** 3 good
**Contribution:** 3 good
**Rating:** 7
**Confidence:** 2

**Summary:**

In this paper, topological parallax is introduced to compare a trained model with a dataset and determine if they share similar multiscale geometric structures. The authors argue that the model is "good" if the geometries are similar. To determine the similarity between the model and the data, they calculate a homologous matching that can be applied to many ML systems. This method can thus be used to assess whether a model has good generalization or is robust to perturbations. The authors validate topological parallax with a toy data set, and the qualitative and numerical results support the authors' claim.

**Strengths:**

Topological data analysis is emerging as a powerful tool for understanding AI systems. Based on TBA, the proposed topological parallax measures the geometric similarity between model and data, which helps to understand whether the trained model is good or not. Moreover, it has many good properties, e.g., it is stable to perturbation, which is crucial for AI attack detection.

**Weaknesses:**

I think that further experiments could be helpful to understand the proposed tool. For example, Figure 3 shows only the results of one neural network, and it is expected to show the comparison of multiple AI models using topological parallax. For example, compare the decision tree model and the neural network in Figure 1.
Also, try to compare several known neural networks. For example, compare a ReLU MLP with a Sin MLP. Given an image, the ReLU MLP may not fit the data well, but if you replace ReLU with Sin in the MLP (called Sin MLP), the latter method can fit the data with almost zero error.

Sin-MLP, also known as SIREN, was described in the article "Implicit neural representations with periodic activation functions".

The reviewer is not familiar with this area and therefore basically respects the other reviews in principle.

**Questions:**

How can topological parallax be used to derive geometric regularities and improve neural network training?

**Limitations:**

The limitation is well discussed.

---

> ### Author Rebuttal · Authors · 2023-08-09
>
> ### Regarding further experiments
>
> We agree with the reviewer that further exploration of real-world datasets is strongly desired in
> future work. Please see the comments in the general rebuttal for this regard as well as the PDF of attached figures for an example of parallax applied to a model on an imagery dataset.
>
> We thank the reviewer for the suggestion of comparing the decision tree model with the neural network
> of Figure 1, as well as comparison with Sin-MLP and Relu-MLP. The comparison within Figure 1
> would be easy to perform, and importantly, would provide readers with another
> example of parallax and its interpretability. We will put this discussion and analysis in the Supplementary Material.
> The comparisons between Sin-MLP (SIREN) and ReLU-MLP are quite interesting, and we would like to pursue these and additional
> comparisons in followup work, in order
> to keep this submission from being increasingly complex.
>
> As the reviewer noted with Sin-MLP, we may assume that all methods will have "zero" or "almost zero" test error.
> One of the motivating ideas behind parallax is to distinguish models in the cases when traditional metrics (such as
> accuracy) give (near) perfect accuracy and thus are indistinguishable from the lens of the metric. We believe that
> parallax provides an additional metric by quanitifying geometric consistency between model and data.
>
>
> ### Regarding deriving geometric regularities and improving training
>
> Recent advances in topologically-inspired loss functions make including topological
> properties within the loss function feasible (see refs [7, 22, 24] in original submission).
> The workflow laid out in Section 7.1 highlights how parallax can be used in neural network
> training. We hope to implement this technique in code soon and describe its implementation
> and a suite of experiments in future work.

---

### Official Review · Reviewer_kfxD · 2023-07-07

**Soundness:** 4 excellent
**Presentation:** 3 good
**Contribution:** 3 good
**Rating:** 7
**Confidence:** 4

**Summary:**

The paper introduces the concept of parallax as a bi-filtered persistence module that measures the geodesic distortion between the dataset and the model. The paper also proves that parallax is stable under perturbations of the dataset and provides a criterion of homological matching to assess whether the model captures the persistent features of the dataset. The paper demonstrates the effectiveness of parallax on two models using the cyclo-octane dataset and discusses the limitations and future directions of the method.

**Strengths:**

This paper presents an interesting and novel approach to evaluate the geometric similarity between a dataset and a model using topological data analysis. The paper is well-written, clear, and provides sufficient background and motivation for the problem.

**Weaknesses:**

There is little experimental analysis on real-world dataset. Perhaps the authors could consider merging some part into the appendix and include more analysis.

**Questions:**

1. This paper presented a novel theorical way to measure the performance of model by analyzing the geometry of model and data. However, we already have lots of metrics, such as accuracy, which can be measured once the trained and the corresponding dataset are available, it is not clear to me how the proposed method advance the existing method.

2. The paper is more related to math/statistics, while the analysis seems to be very solid. The analysis of overfitting and generalization capability (especially under the setting of covariate shift), which is claimed as the major contribution by the authors, is not easy to follow, more concrete examples are needed for illustration.

3. This paper does not discuss the details of models, i.e., ConvNet, RNN, transformer. I am curious whether the proposed theory could be applied to all models trained by back-propagation?


**Limitations:**

See above

---

> ### Author Rebuttal · Authors · 2023-08-09
>
> ### Regarding real-world datasets
>
> We agree that further exploration of real-world datasets is strongly desired in future work.
> See the comments in the general rebuttal as well as the included PDF for an additional example
> of using parallax to interpret a model of an imagery dataset.
>
> ### Regarding comparison to other methods
>
> We agree that there are a variety of well-studied metrics for understanding the performance of models. We
> believe that parallax can be used in conjunction with other metrics to provide additional
> evidence that a model is well-behaved. For example, it is now common to train models to (near) perfect test
> accuracy, and in such scenarios, metrics like accuracy provide no distinguishability. In fact, this is one of the
> major points made by the Belkin's survey *Fit Without Fear*, which we discuss briefly in the paper but
> are happy to add more detail about.
> Parallax may be used to differentiate and highlight certain models compared to others.
>
> We agree that the connection between parallax and generalization capability is not fully explored in this work.
> Our intention was to introduce the idea of homological matching provided by parallax, prove statements about its stability, and
> show via an example it matches what we believe is a widely held intuition about "the shape" of manifolds under
> the manifold hypothesis. The cyclo-octane example and modified Utah-Teapot example do go a long way, we feel towards demonstrating that perception models
> which satisfy our criterion accept points which are reasonable and reject points which are not, and that the opposite is true
> of perception models which do not satisfy our criterion. This is intuitively connected to the idea of generalization capability,
> however we agree that it is far from a rigorous argument connecting our criterion to precisely quantified statistical
> concepts like the generalization gap. On the other hand, as the *Fantastic Generalization Measures...* survey shows, such
> rigorous arguments are few and far between in the literature. We are planning a new paper with a large scale experimental suite
> attempting to generate such a rigorous argument, at least empirically.
>
> ### Regarding details of models (ConvNet, RNN, transformer, etc)
>
> A fascinating aspect of parallax is that it does not depend whatsoever on the architecture or the training system of the model \\(k\\).
> As long as an evaluation oracle is available, parallax will apply.
>
> Thus, parallax provides a method to compare how well different models match the topology of a dataset,
> without relying on any architectural
> properties of the models.
>
> As discussed in the general rebuttal,
> we do intend to pursue this sort of comparison, particularly for image datasets, in followup work.

---

> ### Comment · Reviewer_kfxD · 2023-08-16
>
> Thanks for the response. I would keep the rating.

---

### Official Review · Reviewer_YL9S · 2023-07-25

**Soundness:** 2 fair
**Presentation:** 1 poor
**Contribution:** 3 good
**Rating:** 5
**Confidence:** 2

**Summary:**

The authors propose a method to evaluate how well a model learned a data distribution based on topological data analysis. The authors assume that the model is a classifier (the output of the model is binary, or consists of a finite set of classes that can be evaluated separately), and topologically compare the set of positive data samples to the set of data points where the model outputs a positive value. Topological properties are computed using filtrations of the Rips complex on the data points. The quality of a model is evaluated by checking 1) at which scale the complex in the full ambient space starts to diverge from a complex that is restricted to the subspace where the model is positive, and 2) if persistent features of the full complex are also present in the restricted complex. The authors given an example of measuring and comparing the quality of two models applied the a small dataset with known geometry of the data manifold.

**Strengths:**

- The approach for measuring the similarities between the manifold learned by a model, and the manifold implicitly described by a set of data samples seems interesting, and is novel as far as I can tell (although I am not familiar with topological data analysis).
- It seems like it could be useful for finding adversarial examples for a model, or identifying regions where a model does not perform well.
- If this approach would work with real-world data (or if it can be extended to do that), it could applicable quite broadly to evaluations of models that learn a distribution in high-dimensional spaces.

**Weaknesses:**

- The empirical evaluation is not thorough enough. I do not have a good intuition in which situations the proposed measure of model quality would be useful, and I do not think it is obvious from the description of the method (see below for a discussion). Therefore, a more thorough empirical evaluation is needed to show in which situations the measure is useful. Specifically, more datasets should be evaluated, and the metric used to evaluate success could be improved as well: the current metric using bond lengths is specific to the dataset used, and would not be useful for comparing the performance across different datasets. A more useful metric might be to use a dense held-out test set, and compare the model quality predicted by the proposed method to the model quality according to the test set.
- The method is not compared to any alternatives for measuring the quality of the distribution learned by a model, or for finding adversarial examples. For example, what are advantages/disadvantages compared to the standard approach of using a held-out test set? (I can imagine that the proposed method might have advantages if the test set cannot sample the space densely enough, but disadvantages for regions outside the convex hull of the training samples.) Or compared to other methods for finding adversarial examples (there is a large body of literature, a lot of work can be found, for example, by searching for "adversarial examples"  in Google Scholar)? This should be at least discussed in the related work, and the advantages/disadvantages are not clear from a theoretical standpoint, ideally an empirical comparison should be provided.
- It seems like the measure could be less useful for detecting erros of the model (false positives or false negatives) outside the convex hull of the data samples, since the space outside the convex hull is not explored by the Rips complex. But it seems like real-world data might have true positives outside convex hull (i.e. the true data manifold may extend significantly outside the convex hull of the data samples). See details for a discussion.
- The computational complexity of the method is not provided. Ideally the authors should provide the number of evaluations of the model that is typically needed, in addition to the overhead from building the Rips complex and computing the algorithm described in Section 7.
- The exposition is very hard to follow for non-experts in topological data analysis, and additionally some symbols/variables are undefined (if these are common symbols in topological data analysis, I am not familiar with them).

More details:

* More empirical evaluation is needed to understand in which situations the proposed measure gives good results. From the description of the proposed measure alone, it does not seem clear to me in which situations the measure gives a good estimate of model quality. On one hand, I can see how topological properties could be relatively stable descriptors of a data manifold in high-dimensional space, but on the other hand, I do not have a good intuition how the data manifold typically behaves in high-dimensional space. For example in the manifold of natural images or their latent features, I can imagine that the true data manifold could lie significantly outside the convex hull of the data samples, and that it would be hard to capture enough data samples to cover the full data manifold in their convex hull. In that case, a good model would not just be a "thickening" of the manifold given by the data samples (as described by the authors), or even a thickening of their convex hull, but would also include samples significantly outside the convex hull. For example, a dataset like CLEVR may contain images showing a blue sphere at multiple random locations in the image, but some regions of the image may not be covered by the blue sphere in any of the samples; in that case, we would still expect a good classifier of the data manifold to also classify images as positive where the blue sphere is in a location that was not directly observed in the data samples. Thus, it seems to me that a good model needs to extrapolate significantly outside of the convex hull of the data samples. Would the Rips complex constructed on the data samples be able to capture the part of the true data manifold that extends beyond the convex hull of the samples? And would the proposed method therefore be able to handle such cases, which might be quite common in real-world datasets? Arguably the manifold would be better behaved in latent spaces that have a more semantical representation of the data, but this then means that the latent space that the proposed measure is applied to has to be chosen carefully, and it is unclear how to choose the latent space.

* The exposition is missing definitions in several places:

    - In Section 1, there are a few missing definitions:
        - $K^\circ$ is not defined.
		- $\overline{K}$ is not defined. Does this denote the set complement? If so, would this not mean that $K^\circ$ is the complement of $K$, and in that case, why have two different notations for the complement?
		- $K^c$ is not defined. It could also be the set complement, but then there would be three different notations for the complement, so I guess both $K^\circ$ and $\overline{K}$ do not actually denote the set complement.
	- In Section 2, there are several missing definitions that make it hard to follow the exposition:
		- A Rips complex is not defined. Rips complex may be well-known in topological data analysis, but I think that only a small fraction of NeurIPS readers will be experts in topological data analysis, so giving a short definition would be good. Also, even if the Rips complex is known to readers, the arguments to the Rips complex that are used here may need to be defined. The first argument X is quite clear, but second argument $K$ is less clear. Does the second argument restrict the metric used to construct the Rips complex to geodesics in the subspace $K$?
		- A filtration of a Rips complex is not defined.
		- A chain $Y$ is not defined.
		- $B$ is not defined.
		- In Corollary 2.5, $i_*$ is not defined.
* Eq. 2.2 could probably be simplified. Since $K$ is defined as subspace of $V$, $\rho_V(Y) \le \rho_K(Y)$ is true for all $Y \in R$ (according to Lemma 2.1) and does not need to be mentioned explicitly in the definition. Therefore a simpler definition would be: $P(X, K, V)_{\alpha,\epsilon} =$ {$Y \in R\ |\ \rho_K(Y) \le \rho_V(Y) + \epsilon, \rho_K(Y) \le \alpha$}
* in Algorithm 7.5, step 2, should f(p) be k(p) instead?
* On Line 21: I would define $K$ as {$x \in V\ |\ k(x) = 1$}, since $k^{-1}$ is not well defined for non-injective functions.

**Questions:**

A preview of the discussion of the advantages/disadvantages compared to related work might be useful, as wall as a discussion of the issue with evaluating regions outside the convex hull of the data samples.

**Limitations:**

A few limitations have been discussed, including that the authors are not certain if the method would work on more complex real-world data.

---

> ### Author Rebuttal · Authors · 2023-08-09
>
> We thank the reviewer for helping identify sections that may be difficult for readers due to the field-specific jargon. We agree that clarification of notation is extremely important to readability to a broad interdisciplinary audience.
> Please see the discussion of clarifying notation in the general rebuttal.
> Here are some specific changes that may be beneficial to the exposition. In addition we will include
> Figure 1 in the attached PDF to help the reader visually connect our definitions to meaning.
>
> - In Definition 1.1, add prose: We define a model \\(K\\) to be the closure of an open set, colloquially known as a "solid," \\(K= \\overline{K^\\circ}\\). For any finite dataset \\(X\\), we consider the collection \\(\\mathcal{M}(X)\\) of all models for which \\(X\\) is contained in the interior of the model, \\( X \\subset K^\\circ \\).
> - In Definition 1.2, add prose: A void is a bounded open set \\(\\Omega\\) in the complement of a model, \\(\\Omega \\subset K^c\\), such that ...
> - In Section 2, Equation (2.1), we will introduce basic definitions of Rips complexes on subsets and cite standard TDA textbooks.
> - To clarify the notion of a "chain" by adding the parenthetical "(a formal sum of simplices)" and referencing Hatcher's *Algebraic Topology*.
> - Just before line 105, add a sentence: Let \\(B_\\alpha(x)\\) denote the closed geodesic ball in \\(V\\) centered at \\(x\\) of radius \\(\\alpha\\).
> - Note that \\(\\iota_*\\) in Definition 2.5 is defined in the previous sentence; \\(\\iota_*\\) is the homomorphism on homology that is induced by \\(\\iota\\) on complexes.
> To help orient the reader, we would provide a citation to *Algebraic Topology* by Hatcher.
> - We agree that Eq 2.2 can be simplified as stated.  It was written as shown to emphasize the two-sided bound on \\(\rho_K(Y)\\), but either way is acceptable.
> - On line 21, and also line 290, replace \\(K=k^{-1}(1)\\) with \\(K=\{x \\in V : k(x)=1}\\) to avoid any confusion regarding the pre-image notation.
> - Yes, in Algorithm 7.5, step 2, \\( f( p ) \\) should be \\( k( p ) \\).
>
> ### Regarding comparison outside the convex hull
>
> We agree that models with good generalization will necessarily allow extrapolation outside the convex hull of the training set.
> This idea is expressed in three ways in the submission.
>
> First, in Section 1.1, lines 59--79, we comment on the relationship of reference [2] *Learning in High Dimension Always Amounts to Extrapolation* and the related topology.
>
> Second, our definitions of \\(X\\) and \\(K\\) in Definition 1.1 require that the dataset \\(X\\) is contained in the *interior* of the model \\(K\\), so there are necessarily points outside the convex hull of \\(X\\) that \\(K\\) would accept.
>
> However, the essence of parallax is to ask ``what can I detect about \\(K\\) using only \\(X\\)?''
> We do not assume that \\(X\\) is an original training set, only that it is an available dataset.
> If more datapoints were available from an additional source,
> such as a generative model or from additional data collection,
> then those points should also be used for parallax.
>
> Third, recall that the main result is a stability result (Theorem 5.3),
> based on pointwise perturbation (Section 3).
> Thus, it exists to provide confidence that these methods remain consistent, even if the dataset is pushed "outwards" (or any other direction) by a modest amount.
>
> We absolutely agree that semantic interpretation of the latent data manifold
> is an extremely slippery concept,
> and we hope that parallax helps capture one (incomplete) aspect of that relationship.
>
> ### Regarding comparison to other methods
>
> We agree that it would be beneficial to the field to compare parallax to other ways of measuring generalization and robustness.
> The most comprehensive reference here is
> *Fanstastic Generalization Measures and Where to Find Them* by Jiang et al (arxiv id 1912.02178).
>
> As discussed in the general rebuttal, we do plan on a broad survey that applies parallax to a wide variety of datasets and network architectures,
> and that survey we will attempt to compute as many of these as is practical.
>
> We could add a brief conjectural discussion to this submission; however many of the generalization measures available do not have
> topological foundations (instead, information-theoretic or statistical foundations),
> so meaningful comparisons are difficult without a years-long interdisciplinary research program.

---

> > ### Comment · Reviewer_YL9S · 2023-08-15
> >
> > Thanks for the interesting discussion and additional results. Adding the clarifications of the notation to the paper would help a lot.
> >
> > Regarding data outside the convex hull, the fact that the test set would also be used to construct the Rips complex, not only the training set, is a good point (it might be good to mention this explicitly in the paper, maybe as part of a description of how the method would be used in practice). Although it still seems likely to me that the true data manifold would extend significantly beyond the convex hull of these data points, so it seems unclear to me how much the Rips complex could help identify false negatives/positives of the model that are outside the convex hull of the data samples, beyond what a held-out test set already provides. A small perturbation of the data samples would probably not help a lot with this problem either, as the distance of the samples outside the convex hull would likely be much larger than any perturbation can be without introducing too many false positives.
> >
> > The added teapot experiment is appreciated, this is closer to the image domain most readers will be interested in (although something like a small 2D circle at random locations in e.g. the upper half of the image might address extrapolation outside the convex hull a bit more directly). However, it is likely that this overfitted model does not only have false negatives outside the convex hull, but also introduces holes inside the convex hull that the proposed method can detect. It would be more interesting to show that the proposed method can more accurately distinguish between a well-trained model, and an overfitted model than existing measures of model generalization/robustness, like a held-out test set.
> >
> > So I think the effectiveness of the method described in the paper is still quite hard to judge given the lack of comparisons to existing methods and the small set of toy experiments.
> >
> > But the idea of using topological data analysis to evaluate a model seems quite novel and interesting, and might inspire future work. Considering this and the clarifications promised by the authors, I raise my score by one point.

---

### Author Rebuttal · Authors · 2023-08-09

We thank the editors and reviewers for their high-quality work.
It is clear that the reviewers read the submission carefully and thoughtfully, and
that the overall editorial process at NeurIPS is efficient and productive.

Overall, the weaknesses raised by the reviewers fell into two clear categories.

### 1. Exposition of Definitions and Notation

The reviewers were very helpful in identifying sections that may be difficult for readers
due to field-specific jargon.
We agree that expository clarification of the definitions
and notation is extremely important for readability among a broad interdisciplinary audience.
We believe we can address these concerns by inserting some additional explanatory words
(e.g. ``... the interior \\(K^\\circ\\)''), adding a table of symbols in the Supplementary Material,
and by including citations to the standard definitions in the most common topology textbooks (Hatcher's *Algebraic Topology* and Mukres' *Topology, 2nd edition*) to orient the reader.
Specific alterations are suggested in the individual rebuttals.

### 2. More comprehensive experimentation, particularly for imaging applications

We agree that additional applications and examples (particularly in imaging data) are desirable to demonstrate the utility of parallax and to advance the field of topological analysis of ML models.

The comments by the various reviewers are exactly in-line with our overall research agenda.

We are preparing a separate manuscript that surveys imaging data and popular vision networks.
The rich field of convolutional neural networks deserves special attention due to the high extrinsic dimension and counter-intuitive metric geometry.

In preparing the current submission,
we found that discussion of the often counter-intuitive metrics took too much attention away from the important discussion of topological interpretation and stability analysis.
Therefore, we intentionally restricted the scope to an introduction to the core definitions and algorithms, and we demonstrated their meaning on examples that did not rely on convolutional layers.

We look forward to presenting followup work on imaging data in the near future,
but we feel that the present submission must stands on its own as laying structural groundwork for that experimentation and exploration.

However, we believe we will be able to offer one additional example in the
Supplemental Material without distracting too much from the intended scope of
the manuscript. In Figure 2 of the attached figures PDF, we have included an
additional experimental example based on a simplified version of the Utah
teapot dataset consisting of images of teapots with their spouts and handles
removed, essentially images of jars with lids. We construct a simple model
which accepts all test data points but has very poor interpolation properties.
We show that parallax detects this poor interpolation and we highlight 6
interpolated images that the model rejects. In particular, this example
exemplifies the concern of Section 6: "if step (2) yields \\(\\lambda_{lo,X} (K) = 0\\), then \\(K\\) has
voids between every pair of points in \\(X\\), possibly due to under-sampling or over-fitting, and should
not be trusted for any interpolative purpose". Additional description of this
dataset and CNN could be included in the Supplemental Material.

---

### Decision · Program_Chairs · 2023-09-21

**Decision:**

Accept (spotlight)

**Comment:**

The paper introduces a new methodology to compare a pre-trained model to a reference dataset in terms of their topological similarities, with the hope of shedding light on the delicate balance between overfitting and generalization. It received consistently positive reviews by all 6 reviewers, who appreciated the conceptual and technical novelty, the underlying idea, the theoretical analysis, and the overall exposition (although with some concerns due to the required mathematical background). Some drawbacks were identified in the limited experimental evaluation, but the reviewers also acknowledged the theoretical nature of the paper. The rebuttal clarified all the pending concerns, and some of the reviewers renovated their appreciation for the paper by upgrading their ratings. This is a solid contribution that brings new understanding in the geometrical properties of neural models, and is likely to have a good impact on the community. Therefore, we recommend acceptance.